



# The collapse of the Laurentide-Cordilleran ice saddle and early opening of the Mackenzie Valley, Northwest Territories, constrained by [10]Be exposure dating

Benjamin J. Stoker[1*], Martin Margold[1*], John C. Gosse[2], Alan J. Hidy[3], Alistair J. Monteath[4], Joseph M. Young[4], Niall Gandy[5,6], Lauren J. Gregoire[6], Sophie L. Norris[7] and Duane Froese[4*]

[1]Department of Physical Geography and Geoecology, Charles University, Albertov 6, 12843 Praha 2, Czech Republic

[2]Department of Earth Sciences, Dalhousie University, 1355 Oxford Street, Halifax B3H 4R2, Nova Scotia, Canada,

[3]Centre for Accelerator Mass Spectrometry, Lawrence Livermore National Laboratory, 7000 East Avenue, Livermore, CA 94550, USA

[4]Department of Earth and Atmospheric Sciences, 1-26 Earth Sciences Building, University of Alberta, Edmonton T6G 2E3, Alberta, Canada,

[5] Department of Natural and Built Environment, Sheffield Hallam University, Sheffield, United Kingdom, S1 1WB

[6]School of Earth and Environment, University of Leeds, Leeds LS2 9JT, UK

[7] Department of Geography, David Turpin Building, University of Victoria, Victoria, V8P 5C2, British Columbia, Canada

*Corresponding authors

*Correspondence to*: Benjamin James Stoker (stokerb@natur.cuni.cz), Martin Margold (margold@natur.cuni.cz), or Duane Froese (duane@ualberta.ca)

**Abstract.** Deglaciation of the northwestern Laurentide Ice Sheet in the central Mackenzie Valley opened the northern portion of the deglacial Ice-Free Corridor between the Laurentide and Cordilleran ice sheets and a drainage route to the Arctic Ocean. In addition, ice-sheet saddle collapse in this section of the Laurentide Ice Sheet has been implicated as a mechanism for delivering substantial freshwater influx into the Arctic Ocean on centennial timescales. However, there is little empirical data to constrain the deglaciation chronology in the central Mackenzie Valley where the northern slopes of the ice saddle were located. Here, we present 30 new [10]Be cosmogenic nuclide exposure dates across six sites, including two elevation transects, which constrain the timing and rate of thinning of the Laurentide Ice Sheet from the area. Our new [10]Be dates indicate that the initial deglaciation of the eastern summits of the central Mackenzie Mountains began at ~15.8 ka (17.1 – 14.6 ka), ~1,000 years earlier than previous reconstructions. The main phase of ice-saddle collapse occurred between ~14.9 and 13.2 ka, consistent with numerical modelling simulations, placing this event within the Bølling–Allerød interval (14.6 – 12.9 ka). Our new dates require a revision of ice margin retreat dynamics, with ice retreating more easterly rather than southward along the Mackenzie Valley. In addition, we quantify a total sea level rise contribution from the Cordilleran-Laurentide ice saddle region of ~11.2 m between 16 ka and 13 ka.



## 1.0 Introduction:

The Laurentide Ice Sheet (LIS) was the largest of the Pleistocene Northern Hemisphere ice sheets at the Last Glacial Maximum (LGM; 26 – 19 ka), with a sea level equivalent of between 60 and 90 m (Licciardi et al., 1998; Clark and Mix, 2002; Dyke et

al., 2002; Simms et al., 2019). During the LGM, the NW sector of the LIS reached its all-time maximum extent, coalescing with the Cordilleran Ice Sheet (CIS) and extending along the range fronts of the Mackenzie and Richardson mountains. The LIS subsumed the Mackenzie Valley, altering the drainage systems and blocking plant and animal taxa migration between continental North America and Beringia (Lemmen et al., 1994). During the last deglaciation, the LIS and CIS ice sheets separated, and the Mackenzie Valley opened, allowing the northward drainage of glacial lakes and a route for the exchange of

flora and fauna between North America and unglaciated Beringia (Smith and Fisher, 1993; Teller et al., 2005; Murton et al., 2010; Heintzman et al., 2016; Meachen et al., 2016; Mitchell et al., 2021). The deglacial Ice-Free Corridor (IFC) between the CIS and LIS has been advocated as one of the possible routes taken by early human populations into North America (Johnston, 1933; Antevs, 1935; Goebel et al., 2008; Ives et al., 2013; Froese et al., 2019; Waters, 2019). At the same time, earlier models suggested a lack of coalescence between the LIS and CIS (Johnston, 1933; Antevs, 1935; Mandryk, 1996), and thus a persistent

IFC between these ice sheets. Here, we follow the convention in which the deglacial separation of the two ice masses is still considered an IFC, with the discussion now focused on its timing and availability (Ives et al., 2013; Froese et al., 2019). In addition, rapid ice sheet surface lowering in this region from a collapse of the CIS-LIS saddle has been implicated as a source of meltwater and sea level rise during Meltwater Pulse 1A (Tarasov et al., 2012, Gomez et al., 2015; Gowan et al., 2016; Gregoire et al., 2016).

Despite its relevance to meltwater routing and migration pathways during the last deglaciation, the dynamics and timing of the NW LIS sector remain one of the most poorly constrained of the whole ice sheet (Fig. 1). Current ice sheet chronologies are anchored on about twenty minimum-limiting radiocarbon ages of varying quality in the ~500 km transect in the central Mackenzie Valley (Fig. 1) and early $^{36}$Cl cosmogenic dates (Duk-Rodkin et al., 1996). Recent advances in terrestrial cosmogenic nuclide (TCN) exposure dating have enhanced our ability to constrain the timing of deglaciation and have been

widely applied to date the retreat of the LIS (Gosse and Phillips, 2001; Briner et al., 2006; Dunai, 2010; Balco, 2011). Recently, TCN exposure dating has been used to further constrain the timing of deglaciation for the sparsely dated central-western and southwestern sectors of the LIS (Fig. 1) (Clark et al., 2022; Norris et al., 2022; Reyes et al., 2022). However, the early deglaciation of the NW LIS is yet to be adequately constrained using this approach. Duk-Rodkin et al. (1996) used early TCN exposure dating methods to provide an age indication on the all-time LIS maximum at ca. 30 ka, with a readvance phase at ca.

22 ka. Although these samples were never published in full, so it is not possible to recalculate these ages. The final deglaciation of the LIS from the central Mackenzie Valley occurred perhaps no later than ca. 13.0 cal. ka BP according to minimum-limiting radiocarbon ages (Dyke et al., 2004; Dalton et al., 2020) (Fig. 1).

In this study, we reconstruct the deglaciation of the NW LIS in the central Mackenzie Valley region. We present 30 new $^{10}$Be TCN exposure dates from erratic boulders across six sites along the Mackenzie Valley between 63°N and 65°N. Our TCN





exposure dates cover a range of latitudes and elevations, which enables a dipstick approach to quantify ice-sheet thinning along with lateral retreat rates. We integrate these data with the existing radiocarbon constraints in a Bayesian model consistent with regional, chronological deglaciation information. We also combine our revised ice-sheet chronology with ice-sheet model outputs from Gregoire et al. (2016) to quantify total ice volume loss and freshwater flux to the Arctic Ocean from a collapsing NW LIS during early deglaciation.

## 2.0 Methods:

### 2.1. TCN exposure dating

#### 2.1.1. Site selection and sampling

We sampled 30 glacial erratic boulders of granite, sandstone and quartzite lithologies from six different sites that span the LGM position of the NW LIS and the eastern section of the central Mackenzie Valley (Fig. 2, Table 1). We chose two sites within the Mackenzie Mountains to date the start of deglaciation. At the Dark Rock Creek site (~1,375 m asl), we targeted a series of lateral meltwater channels incised by meltwater from the LIS as it extended up the Redstone River Valley to Dark Rock Creek at the local LGM (Fig. 2; Fig. 3C). At the Katherine Creek site, we collected samples from a ridgeline (~1,050 m asl) above Katherine Creek at the approximate LGM position (Fig. 3E) (Duk-Rodkin and Hughes, 1991). This site is slightly below the elevation limit of erratic boulder occurrence and the likely true local LGM limit, which has been established as between 1,160 and 1,200 m asl (Duk-Rodin and Hughes, 1991). The remaining four sites were in the eastern portions of the Mackenzie Valley at varying latitudes between ~63°N and ~65°N and at a range of elevations so that we could use the $^{10}$Be dipstick approach to quantify the rate of ice sheet thinning (e.g. Koester et al., 2017; Small et al., 2019). We use the dipstick approach at two sites. First, at 65°N, ~ 65 km east of the Katherine Creek site, we collected samples at the summit of the Discovery Ridge of the Norman Range (~920 m asl) and on the adjacent Mackenzie Valley floor (~200 m asl) (Fig. 3F). Second, at 63°N, we collected ten samples transecting Cap Mountain, the highest peak of the Franklin Mountains (1577 m; samples ranging in elevation from 1,441 – 1,241 m). Additionally, two pairs of samples at the summit of the Smith and Bell ridges of the Franklin Mountains, ca 20 and 40 km south of Cap Mountain, around 800 m asl (Fig. 3A and 3B) were collected. We collected samples using a diamond blade cutoff saw and a hammer and chisel. At each site, we sampled a minimum of three erratic boulders. Boulder sampling followed a standard procedure which aimed to reduce the possibility of prior or incomplete surface exposure (Applegate and Alley, 2011; Balco, 2011; Balco, 2020). In particular, we preferentially sampled the surface of erratic boulders which were: large and well exposed above the ground surface (Heyman et al., 2016), situated on stable ground away from steep slopes (Heyman et al., 2011), display a rounded shape which suggests a longer transport history by the ice sheet, and exhibited limited evidence of surface weathering (Balco, 2011).



### 2.1.2. TCN sample preparation, analysis and measurement

The samples were prepared as BeO targets at the CRISDal Lab, Dalhousie University. To concentrate sufficient (20 g) of quartz from each sample, the following procedure was used. The samples were cleaned, crushed, and ground, and the 250-355 μm fraction was rinsed, leached in aqua regia (2 hours), and etched in HF, before mineral separation using combinations of froth flotation, Frantz magnetic separation, air abrasion, heavy liquids, and controlled digestions of non-quartz phases using hydrofluoric or hexafluorosilicic acids. When the quartz concentration was sufficiently pure (as determined optically and with <100 ppm Al and Ti as determined on a 1 g aliquot with the lab's ICP-OES), approximately 35 wt% of the dried quartz concentrate was removed in an ultrasonic bath with dilute HF as per Kohl and Nishiizumi (1992). The samples were spiked with approximately 240 μg of beryllium from the BeCl2 carrier ('Be-Carrier-31-28Sept2012'; prepared from a deeply mined Ural Mountain phenacite with $^{10}Be/^{9}Be$ below 1 x 10-16), and were digested in a HF-HNO$_3$ mixture, evaporated twice in perchloric acid, and treated with anion and cation column chemistry to isolate the Be2+. After acidifying with perchloric and nitric acid to remove residual B, Be(OH)$_2$ was precipitated using ultrapure ammonia gas, transferred to a cleaned boron-free quartz vial and carefully calcined in a Bunsen burner flame to a white oxide for over three minutes. The BeO's were powdered, mixed 2:3 by volume with high purity niobium powder (325 mesh), and packed into stainless steel cathodes for $^{10}Be/^{9}Be$ measurement at the Center for Accelerator Mass Spectrometry, Lawrence Livermore National Lab (CAMS-LLNL). These measurements were made against the 07KNSTD3110 standard with a known ratio of $^{10}Be/^{9}Be=2850x10^{-15}$ (Nishiizumi et al. 2007). Process blanks were also analysed and used to subtract $^{10}Be$ introduced during target preparation and analysis. For all samples, this correction was less than 1% of the adjusted $^{10}Be$ values.

### 2.1.3. Exposure age calculation

Exposure ages were calculated using the online calculator by Balco et al. (2008; version 3.0; constants 3.0.3) and are reported here using the time-dependent CRONUS LSDn production rate scaling of Lal (1991) and Stone (2000), using the 'primary' calibration dataset of Borchers et al. (2016). Individual ages are reported to one significant figure with a 1σ external error (Balco et al., 2008) and internal error which includes AMS precision, uncertainty in carrier and sample Be concentration as measured by ICP-OES, and error contributed by uncertainty in the process blank. This approach differs from Reyes et al. (2022) and Clark et al. (2022) which used the Arctic production rate (Young et al., 2013) and the Lal/Stone scaling method (Balco, 2008). For comparison, we calculate our ages with both production rates, scaling factors, and glacial isostatic adjustment (GIA) corrections in the supplementary materials (Supplementary table S3, S4, S5, S6).

Boulder surface erosion, snow and vegetation cover, atmospheric mass distribution variations during glacier-interglacial transitions, and elevation changes from GIA can all influence TCN production rates, and therefore need to be considered when interpreting TCN exposure ages. The sampled coarse granitoid surfaces displayed grain-scale relief suggestive of surface erosion by grusification. We selected boulders without distinct weathering rinds, gnammas, rillen, or grus on the surrounding ground. Therefore, we assume surface weathering was limited to a few mm over the exposure period given the dry continental





climate and lack of vegetation on the sampled surfaces. The region has low winter precipitation (161 cm average annual snowfall between 1981 and 2010, with average snow depth not exceeding 30cm) (Government of Canada, 2019), and strong winds in the summit areas are expected to keep the top surfaces of erratics free of snow for most of the winter. Most of the

sampling sites were situated above the tree line, with two sites covered by low density boreal forest which would account for less than a 1% decrease in production rate (Plug et al., 2007). As most of the exposure ages post-date the LGM, the influence of katabatic winds and other atmospheric dynamics changes associated with the LIS and CIS ice sheets would have been brief and therefore limited (Staiger et al., 2007). Previous studies that have investigated the impact of changes in atmospheric mass distribution have found that it results in a younger exposure age calculation, but the impact is minimal (Cuzzone et al., 2016;

Ullman et al., 2016; Dulfer et al., 2021). Coupled with the absence of any model at a suitable resolution, we choose not to make corrections for changes in the atmosphere. We therefore do not adjust TCN production rates for erosion, snow or vegetation cover, or atmospheric changes during exposure as any effects are likely to be minimal effect and have large uncertainties. As more research on local temporal variations in atmospheric dynamics in deglaciating regions provides more insight, recalculations of these (and other) TCN exposure ages may be necessary (Jones et al., 2019).

In contrast, the effect of GIA following the deglaciation of the continental ice sheets is reasonably well constrained in the Mackenzie Valley (Peltier et al., 2015; Lambeck et al., 2017; Gowan et al., 2021). A correction is needed to account for how atmospheric shielding varies as boulders are raised from lower elevations to their present day elevations (Jones et al., 2017). The magnitude of GIA varies among our sampling sites, so a correction for GIA-induced change is important to ensure an internally consistent dataset (Fig. S1). In addition, recent studies in adjacent regions of the LIS also included a GIA correction

(Norris et al., 2022; Reyes et al., 2022) and hence, to ensure comparability with these datasets, we apply a correction for GIA related changes to our dataset as well. The previously mentioned changes in atmospheric conditions following deglaciation work against the impact of GIA on calculated exposure ages, but the impact on exposure ages is likely an order of magnitude lower than that of GIA induced impacts (Staiger et al., 2007; Cuzzone et al., 2016; Ullman et al., 2016; Dulfer et al., 2021). We calculated the GIA correction following the method of Norris et al. (2022). We first performed a sensitivity test to

determine the impact of different GIA models on the exposure ages in Octave v.6.4.0 using the Expage-201912 calculator, an open-source script based on the equations of the CRONUS calculator (Table S1 and Fig. S1). Following this, we selected the Lambeck et al. (2017) model as the most suitable for calculating GIA corrections due to the higher model resolution (0.25×0.25 degree). First, we identified when a site became ice-free according to the model of Lambeck et al. (2017). Then, we extracted the change in elevation relative to sea level (Δ RSL) data of Lambeck et al. (2017) for each site at 500-year timesteps, this

allowed us to calculate an average Δ RSL for the time since deglaciation. We then corrected the modern elevation of each sample by the average Δ RSL for the site, resulting in an average site elevation since deglaciation. This GIA correction makes the ages between 0.1% and 3.5% older depending on the site and GIA history (Table 1).



### 2.1.4 Bayesian age modelling

To reduce the uncertainties of exposure ages and support outlier identification we combined our new TCN exposure dates
within Bayesian chronologies using Oxcal v.4.4 (Bronk Ramsey, 2017). These chronologies followed an age-elevation prior
model (Buck et al., 1996), that assumes higher elevations were deglaciated before lower elevations, and therefore accurate
exposure dates from these sites must be older than exposure dates from lower elevations (e.g. Jones et al., 2015; Small et al.,
2020).

Following this prior model, we developed two uniform phase models that included groups of TCN exposure ages and
minimum-limiting radiocarbon dates within sequential phases (Bronk Ramsey, 2009a). These models were run interactively
using different approaches for outlier detection. Oxcal enables the statistical detection of potentially outlying dates in two main
ways, either using the Agreement Index (a measurement of convergence between the unmodelled and modelled probability
distributions of individual ages or the model as a whole) or the Outlier Model command (an outlier analysis that progressively
down-weighs spurious dates) (Bronk Ramsey 2009b). In each case, the first model iteration was run without the outlier function
in order to assess overall and individual Agreement Indices. If the overall Agreement Index of the model was below the 60%
threshold suggested by Bronk Ramsey (2009b), TCN exposure dates with an individual Agreement Index of <60% were
considered for rejection in subsequent model runs. The next model iteration was run using a general outlier model which uses
a student's t-distribution on a timescale of 1-10,000 years (Bronk Ramsey 2009b). In this model, each TCN exposure date was
assigned 0.1 prior probability of being an outlier (i.e. 1 in 10) while flagging potentially outlying dates. Radiocarbon dates
were assigned a 0.05 prior probability of being an outlier (i.e. 1 in 20) and inserted into the models using the *Before* function
which considers dates as minimum (*terminus ante quem*) age controls only (Bronk Ramsey, 2009b).  The final model iteration
was run using the same outlier model parameters as the second model iteration; however, with the exclusion of TCN exposure
dates that fell substantially below the 60% Agreement Index threshold in the first model iteration. The Bayesian syntax is
provided in the supplemental materials (Fig. S7 and S8).

### 2.2 Compilation of existing chronological constraints:

Radiocarbon dating of organic material can be used to constrain the timing of deglaciation although it is subject to uncertainties
inherent to the method. The time lag between deglaciation and the accumulation of organic material means that radiocarbon
ages can only serve as a minimum constraint on the timing of deglaciation. We selected all ages relating to the deglaciation of
the NW LIS from the database of Dalton et al. (2020). We recalibrate the radiocarbon dates using the online calibration tool
Calib v8.2 with the updated IntCal20 northern hemisphere calibration curve (Reimer et al., 2020; Stuiver et al., 2021). We
provide an updated version of the Dalton et al. (2020) database (Table S3) and refer to the median calibrated ages for the rest
of this discussion.

Similar to radiocarbon dating, luminescence dating from post-glacial sites can provide a minimum age on when an area became
ice-free. We compile previously published luminescence ages (optically stimulated luminescence, OSL; and infrared



stimulated luminescence, IRSL); deglacial luminescence ages within the region are only located in the north (Bateman and
Murton, 2006; Murton et al., 2007; 2010; 2015) (Fig. 2). We manually select the oldest dates when building our chronology,
as they are most relevant to reconstructing the timing of deglaciation. Post-glacial features (e.g. dunes) have also been
previously dated with OSL outside of our study area on the bed of the former W LIS (Wolfe et al., 2004; 2006; 2007; Munyikwa
et al., 2011); we refer to these in the discussion but do not include them in Table S2.

Surface exposure dating allows for direct dating of glacial erratics and sediments, without the time lags inherent to radiocarbon
dating. No TCN exposure dates have been fully published that would date the deglaciation within our study area. Early work
in TCN exposure dating by Duk-Rodkin et al. (1996) present a single figure with $^{36}$Cl exposure ages, ranging from 37 ka to 15
ka, which constrain the LIS limits in the Mackenzie Mountains to the Late Wisconsinan (Marine Isotope Stage 2). However,
the calculated ages and the full details of sample analysis were not published for these samples, which are therefore not possible

to recalculate. In addition, our understanding of the calculation of TCN exposure ages has advanced considerably since these
ages were published. As a result, these ages are not comparable to our data in their original form, and so do not refer to them
any further.

**2.3 Modelling ice-sheet saddle collapse:**

We use the ice sheet model simulations of Gregoire et al. (2016) to derive information on plausible ice sheet evolution in the

CIS-LIS saddle region during deglaciation. Gregoire et al. (2016) ran an ensemble of simulations of the North American Ice
Sheet Complex using the Glimmer-CISM thermodynamic ice sheet model. The model runs span the range of uncertainties in
model input parameters, including the ice sheet flow factor, the geothermal heat flux, the basal sliding parameter, the mantle
relaxation time, positive degree day factors, and the atmospheric lapse rate.

The ice sheet mass balance was calculated using a positive degree day mass balance scheme, forced with output from two

general circulation models (GCMs). Here, we present further analysis of simulations from the "Cano" ensemble of ice sheet
simulations see (Gregoire et al., 2016). These simulations use the Trace 21 ka climate simulations to produce an anomaly
forcing, correcting for the present-day model biases in the climatology of the GCM CCSM3. The deglacial forcing for Cano
is calculated in this way:

$$T_{ano}(x, y, t) = T(x, y, t) - \left( \left( T_{PD}(x, y, m(t)) + \gamma H_{PD}(x, y, m(t)) \right) - \left( T_{clim}(x, y, m(t)) + \gamma H_{clim}(x, y, m(t)) \right) \right)$$


$$P_{ano}(x, y, m, t) = P(x, y, m) \times \frac{P_{PD}(x, y, m(t))}{P_{clim}(x, y, m(t))}$$

where T is surface temperature (°C); P is precipitation (mm day$^{-1}$); H is orographic elevation in the model
(metres) at the Present Day (PD) or corresponding to the reanalysis (clim); γ is the lapse rate (-5 °C /km); x and y
represent the latitude/longitude spatial dimensions (metres), m is an index for the months of the year (1-12; Jan-
Dec) and t is the time (years). This climate forcing simulates warming at 14.7 ka, during the Bølling-Allerød

interval, allowing the ice sheet simulations to span a reasonable deglaciation pattern.
The ensemble produces a large range of results. To only analyse the most realistic simulation, Gregoire et al., (2016) applied
constraints on the ice extent and volume, referring to the resultant subset of simulations as "Not ruled Out Yet. We focus on
these "Not Ruled Out Yet" simulations selected by Gregoire et al. (2016) for matching constraints on ice sheet volume and
area. We apply a further requirement that the Franklin Mountains region should not have deglaciated prior to 16 ka in

agreement with the data presented here, leading to a selection of 14 simulations. For each run in the selected 'Cano' ensemble,



we identify the timing of the peak ice volume loss (Fig. 5) and calculate the equivalent sea-level contribution from the saddle region over a 340-year period centred around the maximum rate of volume loss (Fig 5B), assuming an ocean area of 360,768,600 km$^2$.

## 3.0 Results:

In this section, we briefly describe our sites and report the calculated exposure ages. Detailed information on each sample is provided in Table 1, including exposure age calculations with and without the GIA correction. From here, we refer only to calculated exposure ages corrected for GIA, unless specified otherwise. First, we describe the results of our TCN exposure age calculations and then we provide a description of the results from our Bayesian models.

### 3.1 Dark Rock Creek

At Dark Rock Creek (Fig. 3C) we sampled four large, angular quartzite erratics which appeared to originate from local bedrock (NW-18-15 to NW-18-18; Fig. S2V-Y). These erratics were all located on the raised ground between a series of lateral meltwater channels that were cut at the margin of the LIS (Fig. 3C). The resulting TCN $^{10}$Be exposure ages from Dark Rock Creek are poorly clustered and cannot be used to infer an age for the LIS meltwater channels at this site (Table 1). The age spread and tendency towards older ages at this site suggest a strong TCN signal resulting from prior exposure. As the sampled 240 boulders all appeared to be of local material, they likely experienced short transport distances and minimal erosion leading to inheritance. Due to the un-replicated results of this site, we do not refer to data from Dark Rock Creek further when building our chronology of the LIS deglaciation.

### 3.2 Katherine Creek

At Katherine Creek, we sampled five granite erratics along a ridgeline above the western Mackenzie Valley (NW-18-10 to 245 NW-18-14; Fig. 3E; Fig. S2S-U). Two of these samples could not be analysed due to insufficient quartz content (NW-18-13 and NW-18-14). The samples were all collected from within 3 kilometres of each other and at a similar elevation (1,048 – 1,075 m asl). They resulted in exposure ages of 16.7 ± 1.4 ka (NW-18-10), 16.0 ± 1.4 ka (NW-18-11), and 16.1 ± 1.4 ka (NW-18-12). This site is located near the ice sheet margin so there is little GIA-related elevation change (Fig. S1) and minimal impacts on our calculated exposure ages (Table 1; Fig. 6B).

### 3.3 Norman Range and the Mackenzie Valley

The Norman Range forms the northernmost extension of the Franklin Mountains, with approximately 700 m of relative relief over the adjacent eastern Mackenzie Valley (Fig. 3D and 3F). We sampled three granite erratics from the summit (NW-18-07 to NW-18-09; Fig. S2Q-R; Table 1; Fig. 6B) of Discovery Ridge. One sample had insufficient quartz to be analysed (NW-18-08). The remaining two boulders returned exposure ages of 13.3 ± 1.1 ka (943 m asl) and 13.4 ± 1.1 ka (924 m asl). We 255 sampled four erratic boulders from the floor of the Mackenzie Valley ~15 km west of the summit samples (NW-18-19 to NW-





18-22; 192 – 207 m a.s.l.; Fig. S2Z-AC); three of the boulders were composed of granite (NW-18-20 to NW-18-22) and one was pink sandstone (NW-18-19). These samples returned exposure ages of 12.9 ± 1.1 ka (NW-18-19), 14.7 ± 1.3 ka (NW-18-20), 19.2 ± 1.6 ka (NW-18-21), 12.8 ± 1.1 ka (NW-18-22). The influence of GIA at this site is limited to a maximum of 0.2 kyr for our non-outlier samples, due to the close proximity to the LGM ice sheet margin (Fig. S1; Table 1).

### 3.4 Cap Mountain and the lower Franklin Mountains

In the Franklin Mountains, we sampled 14 boulders across two separate elevation groups which occupy distinct elevations (Fig. 3A and 3B). At our upper elevation site, located near the summit of Cap Mountain, we sampled eight granite erratics between 1,279 – 1,479 m asl (NWT-MM-15-01 to NWT-MM-15-08) (Table 1; Fig 6A). The sampled boulders were all highly rounded and rested directly on the local bedrock or, where the bedrock was transitioning into a poorly developed blockfield, lay between the local blocks and boulders. The exposure ages at this site range from 19.2 ± 1.3 ka (NWT-MM-15-01) to 14.4 ± 0.9 ka (NWT-MM-15-07). A further six boulders were sampled across the 'lower Franklin Mountains' group. This includes two boulders from ~2km east of the summit of Cap Mountain, at an elevation of 832 m asl (NWT-MM-15-09) and 814 m asl (NWT-MM-15-10). These returned exposure ages of 14.9 ± 1.1 ka and 15.8 ± 1.1 ka respectively. Two more boulders were sampled from the summit of the Smith Ridge, which is a ridgeline in the Franklin Mountains, approximately 20 km south of Cap Mountain. These boulders were located at 987 m asl (NWT-MM-15-11) and 985 m asl (NWT-MM-15-12) with exposure ages of 15.2 ± 1.0 ka and 13.4 ± 0.9 ka. The final two boulders were sampled on the summit of the Bell Ridge which is located approximately 20 km south of the Smith Ridge. These boulders were located at 854 m asl (NWT-MM-15-13) and 847 m asl (NWT-MM-15-14). The exposure ages were calculated as 14.2 ± 0.9 ka and 13.8 ± 0.9 ka. We acknowledge that there is some distance between the samples of the lower elevation group; however, given the regional ice sheet deglaciation history, the ~40km horizontal distance between these sites is relatively minor and beyond the resolution of TCN exposure dating.

### 3.5 Bayesian models

The first uniform phase model included exposure ages from study sites at Katherine Creek, Norman Range and the Mackenzie Valley, grouped within three sequential phases (Fig. S3). One TCN nuclide date was removed from the model (NW-18-21). This date is several thousand years older than any of the other exposure ages used in the model and could be confidently excluded using either the Oxcal outlier analyses or manual rejection following age-elevation relations. Since the modelled ages all display a normal distribution, we calculate a mean average for each site to provide a best estimate of the timing of deglaciation. Based on our vetted model, our results suggest that Katherine Creek was deglaciated at ca. 15.8 ka (with a two σ maximum-minimum range 17.1 – 14.6 ka). Following this, the summit of the Norman Range (~900m) became ice-free at ca. 14.2 ka (15.0 – 13.6 ka). Finally, the Mackenzie Valley floor deglaciated at ca. 13.6 ka (14.1 – 13.2 ka). Alternatively, a more conservative approach could use the median value of end boundaries between phases for each site to provide a minimum estimate on the timing of ice-free conditions. We provide the information from our Bayesian model outputs in Table S3 and Figure S7. In comparison, the exposure age calculation using the Arctic production rate suggests a mean site deglaciation age





of 17.1 ka (18.2 – 16.1 ka) for Katherine Creek, while the Norman Range deglaciated at 14.9 ka (15.6 – 14.2 ka) and the Mackenzie Valley at 14.2 ka (14.8 – 13.6 ka) (Table S5; Fig. S5).

The second model included exposure ages from study sites at Cap Mountain summit and Lower Franklin Mountains, grouped within two sequential phases (Fig. S4). Within Model iteration 1, two TCN exposure dates (NWT-MM_15-01 and NWT-MM-15-08) had individual Agreement Index values substantially <60% and reduced the overall Agreement Index of the model to <60%. These were rejected from Model iteration 3. Results from this model indicate a mean average timing of deglaciation of ca. 14.9 ka (15.7 – 14.2 ka) for our upper elevation samples near the summit of Cap Mountain (1,400m). The lower elevation

sites (~800m) became ice free around 600 years later, at 14.3 ka (15.1 – 13.6 ka). Full information from our Bayesian model outputs is available in Table S4 and Figure S8. The alternate exposure age calculation results in a mean site deglaciation age of 17.0 ka (18.5 – 15.3 ka) for the Cap Mountain summit site, which is considerably older than our calculation. The calculation of exposure ages using the Arctic Production rates results in lower uncertainties, which means the spread of ages cannot be well constrained by the Bayesian model and results in two clusters of ages for this site (Fig. 8; Table S6; Fig. S6). The Franklin

Mountains site deglaciated at 14.9 ka (15.6 – 14.2 ka) using this calculation method.

## 4.0 Discussion:

### 4.1.1 Implications for the regional retreat pattern

The existing ice margin chronology of Dalton et al. (2020) in the Mackenzie valley largely follows the models of Lemmen et al. (1994) and Dyke et al. (2003) and portrays the eastern ridges of the Mackenzie Mountains at 63°N as glaciated until ~15.5

cal. ka. Following initial retreat, ice persists in the central Mackenzie valley until ~13.5 cal. ka with the NW LIS margin abutting against the slopes of the Mackenzie Mountains (Fig. 4). The Dalton et al. (2020) ice margin chronology is based on minimum-limiting radiocarbon ages, which for this region vary between 11.5 and 9.0 [14]C ka (13.4 and 10.3 cal ka). In contrast, our ages indicate an earlier start of deglaciation and a change in the retreat pattern and style. We present a new chronological model for the deglaciation of the NW LIS in Figure 4, a full justification of our changes to the ice margin pattern is described

in Supplementary Document 1. To allow for a more direct comparison of ice retreat, we use the time slices from the previous reconstructions of Dyke et al. (2003) and Dalton et al. (2020).

The new [10]Be ages indicate deglaciation of the eastern summits of the central Mackenzie Mountains began at ~15.8 ka (17.1 – 14.6 ka) (Fig. 4A). This is compatible with existing data on the timing of advance and retreat of the NW LIS further north in the Richardson Mountains, where radiocarbon ages indicate that the LIS reached its maximum sometime after ~19.1 cal. ka

BP (18.9 – 19.4 cal ka BP) and began to retreat from its maximum position by ~16.6 cal. ka BP (Kennedy et al., 2010; Lacelle et al., 2013). Further north, the LIS reached its short-lived maximum in the Mackenzie Delta region between 16.6 ka and 15.9 ka (Bateman and Murton, 2006; Murton et al., 2007; 2015). These constraints are closely aligned and suggest that the initial deglaciation of the NW LIS at ~16 ka was broadly synchronous from the Mackenzie Delta and Richardson Mountains to the Mackenzie Mountains.



Following the start of deglaciation, the NW LIS margin at 63 - 65 °N remained stable as the ice sheet occupied the Mackenzie Valley with the ice margin pressed against the eastern slopes of the Mackenzie Mountains during a period of ice sheet thinning. The highest summits (~1,400 m) of the southern Franklin Mountains became ice-free at ~14.9 ka (15.7 – 14.2 ka) and its lower elevations (~900 m) by ~14.3 ka (15.1 – 13.6 ka) (Fig. 4B - C). Compared with the reconstruction of Dalton et al. (2020) and Dyke et al. (2003), these constraints suggest an earlier deglaciation of the Franklin Mountains at 63°N by around 1,000 years.

Exposure ages from ~1,000 m elevation in the Rocky Mountains at 58.7°N indicate LIS deglaciation at 14.4 ± 0.6 ka (Clark et al., 2022). OSL ages from dune fields provide a minimum constraint on deglaciation and suggest the separation of the LIS and CIS as early as 15 ka in central Alberta (Munyikwa et al., 2011), with ice sheet retreat to the south of Great Slave Lake likely by ~13.4 ka (Wolfe et al., 2007; High Level, Alberta; Norris et al., 2021) and definitely by 10.5 ka (Wolfe et al., 2004; Sandy Lake, Alberta). These data are all consistent with our new chronological constraints. Similar shifts to an earlier timing of

deglaciation from existing radiocarbon-based chronologies have been suggested by recent TCN exposure dating studies (Norris et al., 2022; Reyes et al., 2022).

At 65°N, ice remained over the Norman Range (900 m asl) until ~14.2 ka (15.0 – 13.6 ka), and at lower elevations (~200 m asl) in the Mackenzie Valley until ~13.6 ka (14.1 – 13.2 ka) (Fig. 4D). This aligns with previous studies which suggested that the Mackenzie Valley must have been ice-free by the Younger Dryas time period (12.9 – 11.6 ka) (Smith, 1994; Gowan, 2013;

Gowan et al., 2016; Dalton et al., 2020). Nearby radiocarbon dates from the Little Bear River delta (~50 km south) suggest that Glacial Lake Mackenzie occupied this area at 13.4 ± 0.17 cal ka BP (13.1 – 13.8 cal ka BP; I-15020). Ice margin retreat continued to the east of our study area, reaching the Canadian Shield at ~115.7°W, by at least 12.6 ka (11.8 – 13.4 ka) (Reyes et al., 2022).

Our new age constraints necessitate a revision of the ice margin retreat pattern and timing. Our TCN exposure ages suggest

that the Franklin Mountains at 63°N were deglaciated down to ~800 m by 14.3 ka, around 1,000 years earlier than in the reconstruction of Dalton et al. (2020). The Norman Range at 65°N and a similar elevation (~900m) deglaciated at broadly the same time, ~14.2 ka, with the Mackenzie Valley at ~200m glaciated until 13.6 ka. The age constraints at 65°N are broadly consistent with the existing radiocarbon reconstruction of Dalton et al. (2020). These changes to the chronology result in a shift in the retreat pattern compared with past reconstructions, which suggested ice retreated to the south, up the Mackenzie

Valley. We find no evidence that the deglaciation of the Mackenzie Valley region around 63°N lagged the timing of deglaciation around 65°N, as suggested by previous reconstructions (Dalton et al., 2020). Instead, we propose that the retreat of the LIS across this region was broadly synchronous and to the east, with the final deglaciation involving topographically constrained ice lobes occupying lower elevations (Fig. 4).

### 4.1.2 A comparison of different approaches to exposure age calculation

The use of different calculation methods or calibration data when calculating TCN exposure ages can result in changes in the exposure age. A range of approaches have been used when calculating exposure ages in recently published datasets for the W LIS, including the use of different production rates, scaling methods, and GIA corrections (cf. Clark et al., 2022; Norris et al.,



2022; Reyes et al., 2022). Issues arise when trying to compare exposure age datasets with different calculation approaches as the published deglaciation ages are often incompatible. Reyes et al (2022) suggest the NW LIS retreated to the Canadian

Shield, to the east of our study area, by 13.9 ± 0.6 ka (Fig. 1), conflicting with our new exposure ages in the Mackenzie Valley region. Reyes et al. (2022) used the Arctic production rate (Young et al., 2012), the Lal/Stone scaling method (Balco et al., 2008), and a modelled GIA correction in their exposure age calculations. In Table 1, we provide exposure ages calculated following our preferred method and following the method detailed in Reyes et al (2022). The GIA correction of Reyes et al. (2022) is based on simulations from Tarasov et al. (2012) which are not available, therefore we cannot replicate this method

of GIA correction. Instead, we apply the GIA correction described in our methods section and following the method of Norris et al. (2022). Our GIA correction results in exposure ages which are ~3.5 – 4% younger, for the sites presented in Reyes et al. (2022).

The choice of production rate has the strongest influence on the calculated exposure age, with the scaling factor having a lesser effect. Exposure ages calculated using the Arctic production rate (Young et al., 2013) instead of the 'primary' calibration

dataset of Borchers et al. (2015) are ~8.5% older. The choice of scaling method only has ~2% influence on the calculated exposure age. The Arctic production rate calibration data set is composed of eighteen samples from five sites in west Greenland and east Baffin Island (Young et al., 2013). These samples occupy a narrow latitudinal range (69.1°N – 69.8°N), low elevations (65 – 350m), and are relatively young (8.2 – 9.2 ka). While our sites are situated in the Subarctic (63°N - 65°N), the other characteristics of our sites make it inappropriate to use the Arctic production rate calibration data set to calculate our exposure

ages. Most importantly, multiple sites are situated at or around 1,000m elevation and may be as old as 16 – 17 ka. The 'primary' calibration data set of Borchers et al. (2015) includes 47 samples at five sites which cover a range of elevations, ages, and latitudes which we think are more representative of the range of sites in this study.

The 'primary' calibration data set (Borchers et al., 2015) produces results which are compatible with the existing chronological constraints on the deglaciation of the NW LIS (Fig. 7). The NW LIS was advancing to the Richardson Mountains around 18.6

ka (Kennedy et al., 2010; Lacelle et al., 2013), and to the Mackenzie Delta possibly as late as 16.6 ± 0.9 ka (Murton et al., 2015). The local LGM was short-lived, with retreat beginning by ~16 ka in the Richardson Mountains and Mackenzie Delta (Bateman and Murton, 2006; Kennedy et al., 2010; Murton et al., 2015). The eastern peaks of the Mackenzie Mountains, around Katherine Creek, should deglaciate shortly after this. While Cap Mountain, in the Franklin Mountains, should deglaciate even later. However, exposure ages calculated with the Arctic production rate are in conflict with this reasoning, as

they suggest deglaciation at Katherine Creek began at as early as 17.1 ka (18.2 – 16.1 ka) with Cap Mountain ice-free at approximately the same time (17.0 ka; 18.5 – 15.3 ka). Therefore, we favour the 'primary' calibration dataset of Borchers et al. (2015) for calculating the exposure ages in the study as it results in a deglacial model which is consistent with existing geomorphological constraints and its anchored data is representative of our sample sites.

Regardless of the chosen calibration dataset, we are able to make two important contributions to the regional deglaciation

chronology. First, irrespective of production rate, there is a robust trend indicating much earlier deglaciation than has been previously considered for the central Mackenzie valley (Dyke et al., 2003; Dalton et al., 2020). Using the Arctic production





rate (Young et al., 2013) produces a deglacial chronology that is largely too old when compared to the existing age constraints (Fig. 8). While using the 'primary' production rate (Borchers et al., 2015) produces mean site ages which are ~8% younger than the Arctic production rate dates, resulting in a deglacial chronology that is more compatible with the existing
geomorphological constraints (Fig. 7 and 8). Second, both production rates produce a rapidly thinning Laurentide Ice Sheet in the study area, with thinning rates beyond the resolution of TCN exposure dating (Fig. 7 and 8). In both models, ice sheet thinning starts before and continues through the Bølling-Allerød interval in agreement with the model simulations of Gregoire et al. (2016). The majority of ice sheet thinning occurs within the Bølling-Allerød interval when using the 'primary' production rate, whereas a substantial amount of melt occurs before the Bølling-Allerød when using the Arctic production rate.

## 4.2 Implications of ice free Mackenzie Valley for species migration and meltwater routing

### 4.2.1 Opening of the Ice-Free Corridor and faunal migration

The exact timing of the opening of the IFC is contentious. Depending on the chosen chronological constraints and degree of scrutiny of the available data, it has been suggested the IFC may have been viable as early as 14.9 ka, or as late 12.6 ka (Heintzman et al., 2016; Pedersen et al., 2016; Potter et al., 2018; Froese et al., 2019). A primary migration route along the
west coast of North America, termed the Pacific Coastal Route, has been suggested as an alternative to the IFC for early humans (Fladmark, 1979; Braje et al., 2017; Lesnek et al., 2018; Braje et al., 2020). Our reconstruction of the deglaciation of the Mackenzie Valley provides maximum age constraints on the timing of a viable migration route through the northern IFC. Initial separation of the CIS and LIS over the southern Mackenzie Mountains is not chronologically constrained, but further south (55°N), mountain summits over 2000 m asl became ice free as early as 15.6 ± 0.6 ka (Dulfer et al., 2021), consistent
with our earliest ages on the deglaciation in the study area at Katherine Creek at 15.8 ka (17.1 – 14 6 ka). The western margin of the LIS then stayed pressed against the eastern slopes of the Mackenzie Mountains during a period of ice sheet thinning between 14.9 and 14.3 ka (Fig. 4B and 4C). After deglaciation of the Franklin Mountains east of the Mackenzie River (~14.3 in the south and ~13.9 in the north) ice remained in the Mackenzie Valley until about 13.6 ka (14.1 – 13.2 ka; Fig. 4D). Our constraints are incompatible with an early opening of the IFC at around 15.0 cal ka (Potter et al, 2018). Instead, migration
through the Mackenzie Valley portion of the IFC was possible only after ~13.6 ka (Fig. S3). Our constraints for the northern opening of the IFC are in good agreement with those from the southern IFC, which Clark et al. (2022) suggest was fully opened by 13.8 ± 0.5 ka, based on an inventory of 64 [10]Be samples. This chronology is consistent with dating of the arrival of northern (Beringian) bison into Alberta and NE British Columbia through the central and southern IFC by at least 13.2 ka (Froese et al., 2019; Heintzman et al., 2016).

### 4.2.2 Glacial lake drainage

A series of glacial lakes formed along the retreating margin of the LIS, dammed by regional ice retreat and the isostatic depression of topography. The northward drainage of these lakes may have resulted in large fluxes of freshwater to the Arctic





Ocean which could have affected past climate through the disruption of ocean circulation, most notably the Younger Dryas

cold period (12.9 – 11.7 ka) (Broecker et al., 1989; Clark et al, 2001; Teller et al., 2002; Norris et al., 2021). The drainage

route is an important factor in determining the extent to which freshwater fluxes may have disrupted ocean circulation

(Condron and Winsor, 2012; Pendleton et al., 2021). However, our understanding of when certain drainage routes were active

remains poor, making it difficult to examine the links between flood events and ocean/ climate records.

Our TCN exposure ages indicate a viable NW drainage route from glacial Lake Agassiz through the Mackenzie Valley prior

to or at the beginning of the Younger Dryas climate event. Our reconstruction shows the Mackenzie Valley ice-free and

occupied by glacial Lake Mackenzie at ~13.6 ka, which is consistent with existing radiocarbon constraints in the Lake

Makenzie basin (Smith, 1992; Gowan, 2013). Recent work has also identified that the northwestern outlet of glacial Lake

Agassiz was ice free prior to ~13.0 ka (Norris et al., 2022). In addition, flood deposits in the Mackenzie Delta have been dated

to around 13.0 ka (Murton et al., 2010), apparently coincident with a large influx of freshwater into the Beaufort Sea (Keigwin

et al., 2018). However, the specific drainage histories of glacial Lake Agassiz and the intervening glacial lakes along the ~2000

km flood route to the Arctic Ocean are not fully resolved near the onset of the Younger Dryas climate event (Smith, 1992;

Lemmen et al., 1994; Fisher et al., 2008; Young et al., 2021).

### 4.3 Ice sheet thinning and contributions to past sea level rise:

Meltwater Pulse 1A (MWP-1A) represents a period of rapid sea level rise; in ~300 years sea level rose globally by ~15 m

(Deschamps et al., 2012; Church et al., 2013). The timing of this event (14.7 – 14.3 cal ka; Deschamps et al., 2012) coincides

with the abrupt warming during the Bølling-Allerød time-period and its source has been attributed to the rapid melting or

collapse of one or multiple ice sheets. The North American Ice Sheet Complex has been suggested as a substantial contributor

to MWP-1A from models of the distribution of released meltwater based on far field sea level records (Gomez et al., 2015;

Liu et al., 2016; Lin et al., 2021). Numerical modelling indicates that a substantial portion of the meltwater may have originated

from the W LIS, owing to the rapid collapse of an ice saddle between the LIS and the CIS (Fig. 5; Gregoire et al., 2012;

Gregoire et al., 2016; Gowan et al., 2016).

Our TCN exposure ages represent the first direct evidence of ice sheet thinning in the NW portion of the LIS sector. These

observations of ice sheet thinning, which began shortly before, and continued through the Bølling-Allerød interval, support

the ice sheet saddle collapse hypothesis of Gregoire et al. (2016) that abrupt warming triggered the CIS-LIS ice saddle collapse

(Fig. 5D, E; Fig. 6). This chain of events fits with results from the Gregoire et al. (2016) ensemble of model simulations.

Indeed, model results suggest that the CIS-LIS ice saddle collapse is triggered by the abrupt warming only if the saddle

experienced an initial lowering prior to the event (as shown in Fig. 5E). The five simulations with the earliest saddle collapse

experienced an ice lowering of 116-157 m prior to the abrupt warming, whereas the 5 simulations with the latest saddle collapse

had a prior lowering of only 19-47 m (Fig. 5A). This initial lowering was caused by the progressive increases in summer

insulation and greenhouse gases since the LGM (Gregoire et al., 2015). The uncertainties involved in TCN exposure dating

are too large to calculate a precise rate of ice sheet surface lowering for our sites. However, the close alignment of our empirical





ice sheet reconstruction with the model simulations of Gregoire et al (2016) mean that we can quantify the sea level contributions from the collapsing saddle region of the W LIS during the last deglaciation.

Ice sheet modelling of Gregoire et al. (2016) indicates a total of 11.2 m sea level rise contribution during the deglaciation our of study area from 16 – 13 ka, based on the average of 14 simulations. Rapid ice sheet melting from the saddle region is
observed during MWP-1A, resulting in a 3.4 m sea level rise contribution in 340 years. The North American Ice Sheet Complex as a whole contributed 5–6 m or more and the northern slopes of the CIS-LIS ice saddle experienced their most intensive melting during MWP-1A (Gregoire et al., 2016). This is consistent with numerical modelling by Tarasov et al. (2012) that quantified the NAISC contribution as 'likely between 9.4 and 13.2 m over a 500 year interval' with the W LIS providing the largest share of the meltwater. Field data indicate that regions beyond the saddle area, such as the main portion of the CIS, and
the SE sector of the LIS also experienced substantial drawdown during MWP-1A (Menounos et al., 2017; Corbett et al., 2019); this makes the North American Ice Sheet Complex a major contributor to this rapid sea level rise event. The massive meltwater discharge into the Arctic from the ice saddle collapse may have slowed ocean circulation and could have initiated the end of the Bølling-Allerød warming (Ivanovic et al., 2017) with limited counteracting effect from Antarctic meltwater discharge into the Southern Ocean (Ivanovic et al. 2018).

Large marginal spillways incised in along the eastern slopes of Mackenzie Mountains provide evidence for major melting in this part of the LIS (Fig. 9; Duk-Rodkin and Hughes, 1991; Lemmen et al., 1994; Bednarski, 2008). These meltwater channels are oriented parallel to the range front and draining to the north, located near the LGM limits of the NW LIS, and are dated to between 14.9 ka and 14.2 ka by our reconstruction (Fig. 4B and 4C). Thus, they do not record a long-term drainage of meltwater, but rather high-magnitude discharge from the rapidly thinning NW LIS during early retreat.


## 5.0 Conclusions

We provide 30 new [10]Be TCN constraints on the deglaciation of the NW LIS. Bayesian modelling of our TCN exposure ages indicate that the initial deglaciation of the NW LIS in the central Mackenzie Mountains began at ca. 15.8 ka (17.1 – 14.6 ka). The summits (~1,400 m) of the Franklin Mountains (~63°N) deglaciated at ca. 14.9 ka (15.7 – 14.2 ka), with ice sheet surface
lowering down to ~800m by ca. 14.3 ka (15.1 – 13.6 ka). The Norman Range (~900 m) deglaciated at ca. 14.2 ka (15.0 – 13.6 ka) and the adjacent Mackenzie Valley (~200 m) at 65°N deglaciated at ca. 13.6 ka (14.1 – 13.2 ka), opening the northern portion of the IFC. Our revised ice margin retreat chronology of the NW LIS accommodates the available age constraints on deglaciation, including our new TCN exposure ages. The main change in our reconstruction, compared with past work, is an earlier retreat of the NW LIS around 63°N. This results in broadly synchronous deglaciation of the Mackenzie Valley at 63°N
and 65°N, compared with past reconstructions which suggest that the region around 63°N deglaciated ~1 ka later than the region to the north. According to our reconstruction, the IFC was not a viable migration route at the time of the peopling of North America around 14 – 15 ka. We find close agreement between our empirically based reconstruction and the sequence

of events derived from the numerical ice sheet models of Gregoire et al. (2016). We use these model simulations to quantify the sea level contribution of the CIS-LIS saddle region during deglaciation, indicating a cumulative sea level contribution of
11.2 m from 16 – 13 ka.

**Data availability**

The data referred to in this manuscript has all been provided within the tables and figures in the main text and the supplementary materials. If further information is needed, it is available on request from the corresponding authors.

**Supplementary materials**

The supplementary materials related to this article are available online at: https://doi.org/10.6084/m9.figshare.20069222.v1

**Author contributions:**

The project was conceptualized by MM and DF with input from BJS and JMY. Cosmogenic nuclide samples were collected by MM, DF, BJS and JMY. The laboratory processing and analysis of cosmogenic nuclide samples was undertaken by JCG and AJH. BJS curated the data used in this project and completed the analysis. The cosmogenic nuclide exposure age
calculation approach was devised and undertaken by BJS under the supervision of MM and DF. The method to correct exposure ages for glacial isostatic adjustment was developed by SLN and applied to the exposure ages by BJS. The Bayesian model setup was developed by AJM with input from BJS and JMY. The Bayesian model runs were performed by AJM. The ice sheet saddle collapse model was originally developed by LJG and the model analysis presented in this paper was performed by NG. The manuscript was written by BJS and MM with input from all the authors. The data visualisation and figure creation was
completed by BJS with input from all authors.

**Competing interests:**

The authors declare that they have no conflict of interest.

**Acknowledgements:**

We thank Guang Yang of the CRISDal lab at Dalhousie University for the cosmogenic nuclide sample preparation and
chemistry. This research was supported by: the Czech Science Foundation under grant number 19-21216Y and the Swedish Research Council International Postdoctoral Fellowship no. 637-2014-483 awarded to MM, Charles University Grant Agency project GAUK 251363 awarded to BJS. As well as grants from the NRCan Polar Continental Shelf Program and the Natural



Science and Engineering Research Council to DF, and grants from the University of Alberta Northern Research Awards to JMY. DEMs provided by the Polar Geospatial Center under NSF-OPP awards 1043681, 1559691, and 1542736.

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

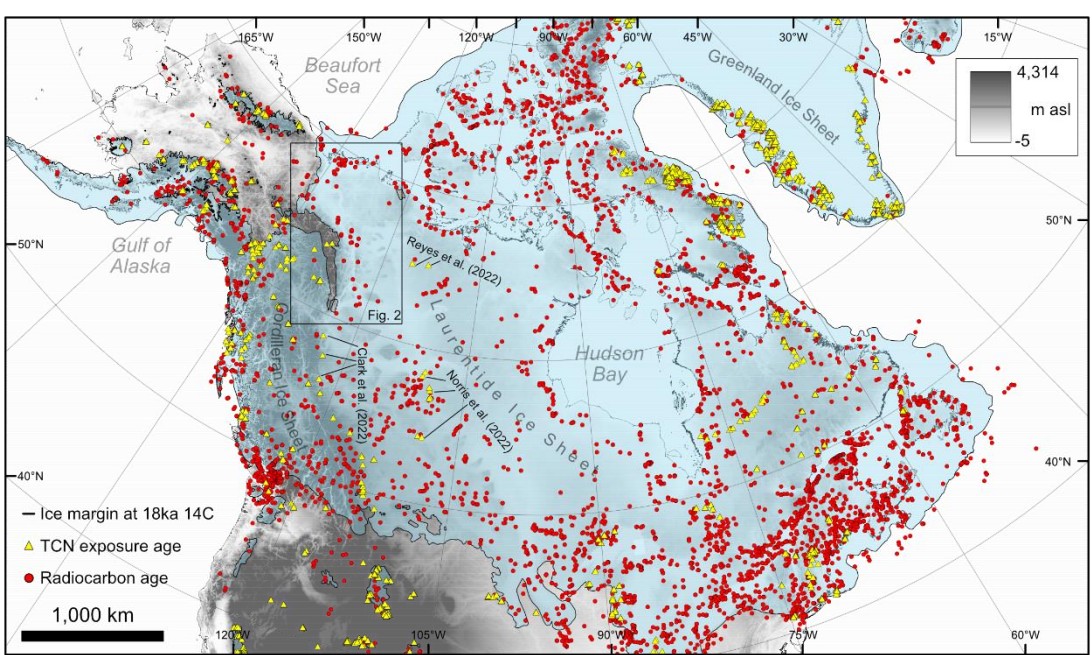

**Figure 1: The Last Glacial Maximum (18.0 $^{14}$C ka/21.1 cal. ka) extent of the North American Ice Sheet Complex according to the reconstruction of Dalton et al. (2020). The yellow triangles (cosmogenic nuclide exposure ages) and red dots (radiocarbon ages) show**
**the distribution of dates constraining deglaciation. The ExPage compilation by Heyman (2022) was used to plot the distribution of cosmogenic nuclide exposure ages and the database of Dalton et al. (2020) was used to plot the distribution of radiocarbon ages.**



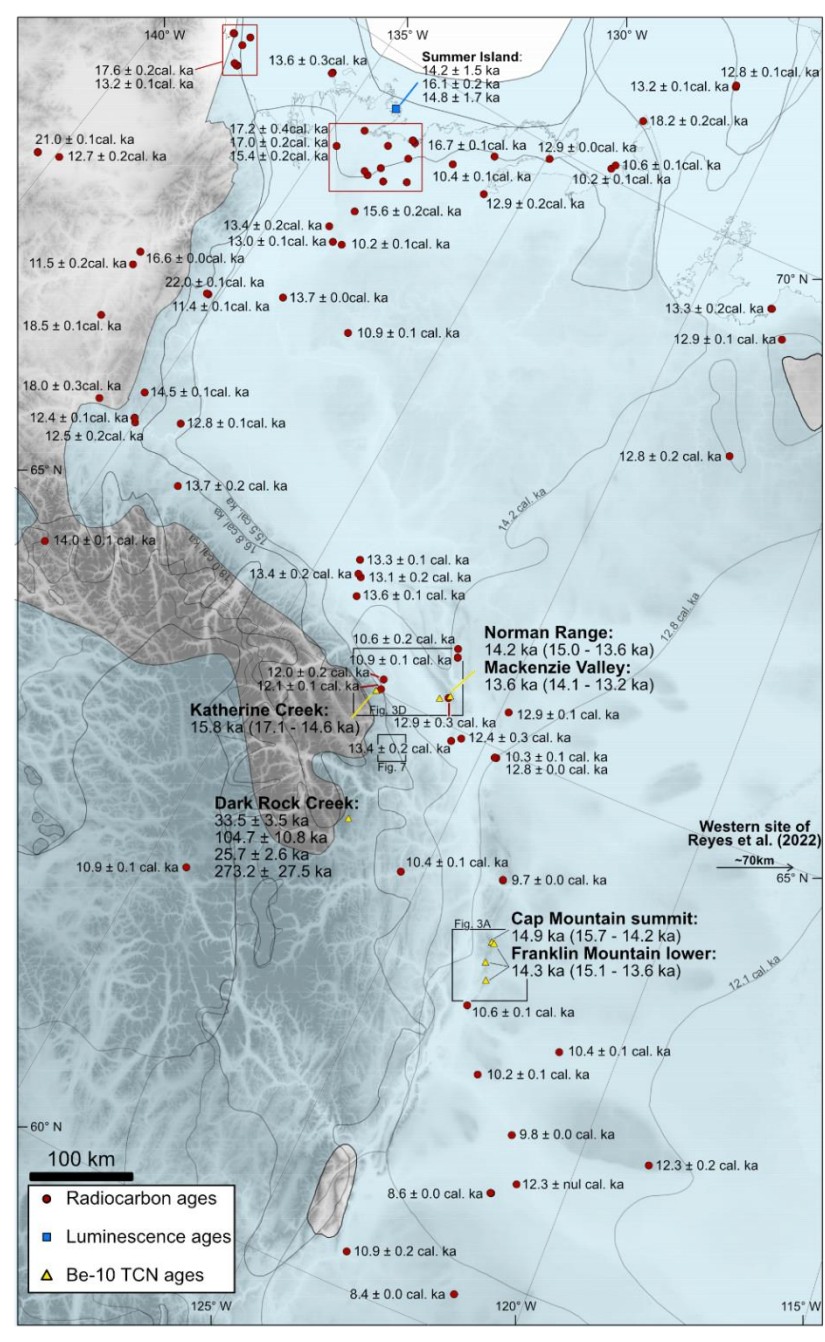

**Figure 2: The distribution of chronological constraints on the deglaciation of the NW sector of the Laurentide Ice Sheet. The pattern of deglaciation is depicted by grey lines and the shaded blue area represents the local LGM limit at 18.0 cal. ka BP (Dalton *et al.*, 2020). For our new TCN exposure ages, we present the mean site deglaciation ages calculated from our Bayesian modelling. Radiocarbon ages were selected from the database of Dalton *et al.* (2020) and recalculated using the IntCal20 curve (Reimer *et al.*, 2020). We only present the oldest OSL ages from post-glacial sediments in the Mackenzie Delta region (Bateman and Murton, 2006; Murton et al., 2007; 2010; 2015).**




**Figure 3: (A)** Location of the sampling sites in the Franklin Mountains. **(B)** Close-up of Cap Mountain, where samples NWT-MM-01 to -10 were collected along an elevation transect. **(C)** Close-up of the Dark Rock Creek site. **(D)** Location of the northern sampling sites. **(E)** Close-up of the Katherine Creek site where samples NWT-18-10 to -12 were collected along the ridge. **(F)** Close-up of the Mackenzie Valley and Norman Range sites where samples NWT-18-19 to -22 and samples NWT-18-07 to -09 were collected. DEM courtesy of the Polar Geospatial Center

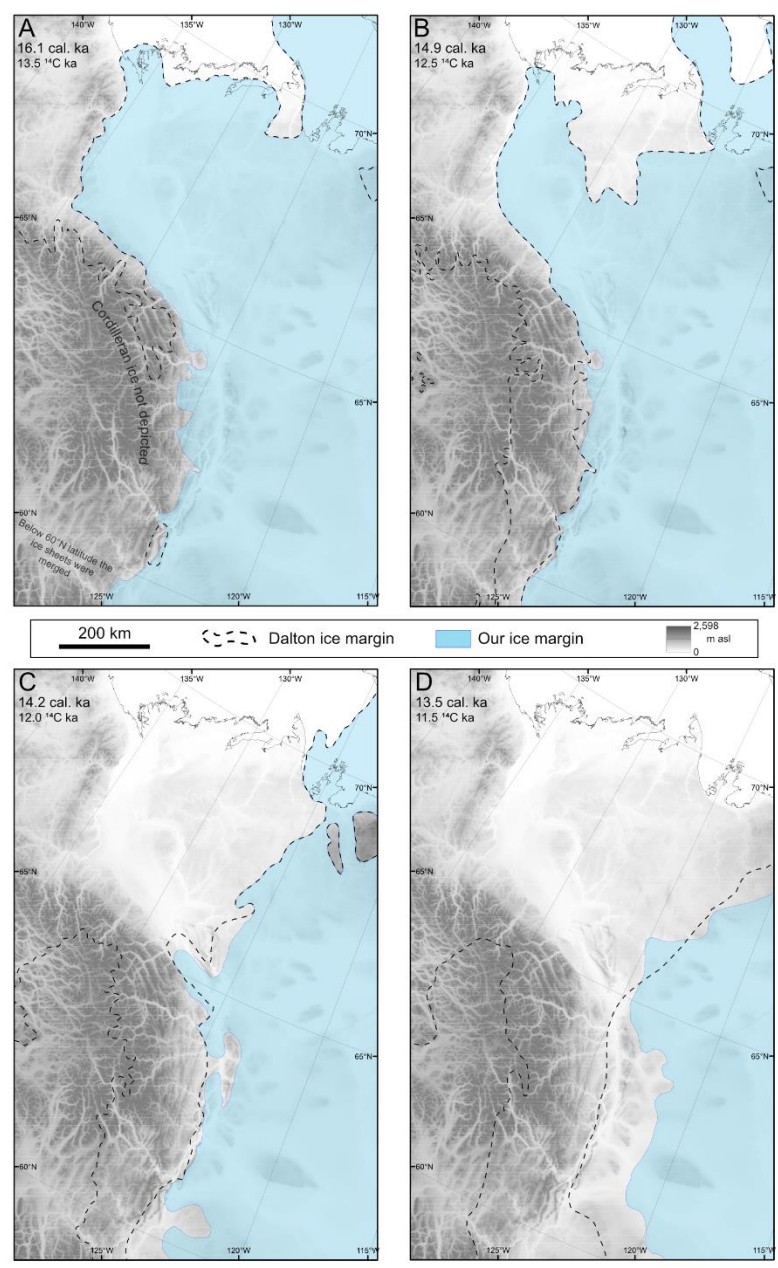

**Figure 4: . Our ice margin retreat chronology compared to the reconstruction of Dalton et al. (2020). We do not depict the Cordilleran Ice Sheet in any of our reconstructions (A) Ice margin reconstruction at 16.1 cal. ka. (B) Ice margin reconstruction at 14.9 cal. ka. (C) Ice margin reconstruction at 14.2 cal. ka. (D) Ice margin reconstruction at 13.5 cal. ka.**






**Figure 5: (A)** Simulated LIS surface elevations at the Franklin Mountains, 18-12 ka, in the 14 selected simulations. **(B)** Sea level flux from the CIS-LIS saddle area. Black points show the point of maximum sea level flux for each simulation. The grey curve shows the sea level flux evolution for one simulation (Cano_34,; peak marked by a cross), with the 340-year period of peak sea level flux in red. **(C)** Ice thickness at 15ka for one simulation (Cano_34), with grey contours every 500 m from 1000 m of ice thickness. X-X' line






**marks the transect used in panel e, the red dot marks the Franklin Mountains where the [10]Be samples were collected, the dashed rectangle marks the CIS-LIS saddle area used for calculations (same in panel d). (D) Surface mass balance (SMB) of the North American Ice Sheet at 16ka (Cano_34). Dark and light purple contours mark equilibrium line at 16 and 14ka respectively, showing the change in SMB that resulted in the CIS-LIS saddle collapse. (E) A transect through the CIS-LIS saddle (Cano_34), at 1000 year intervals from 18 ka (greatest extent) to 12 ka (smallest extent). The black curve shows topography and the grey curves successive ice surfaces, with dashed lines indicating 500 year intervals during the period of most rapid ice sheet thinning.**




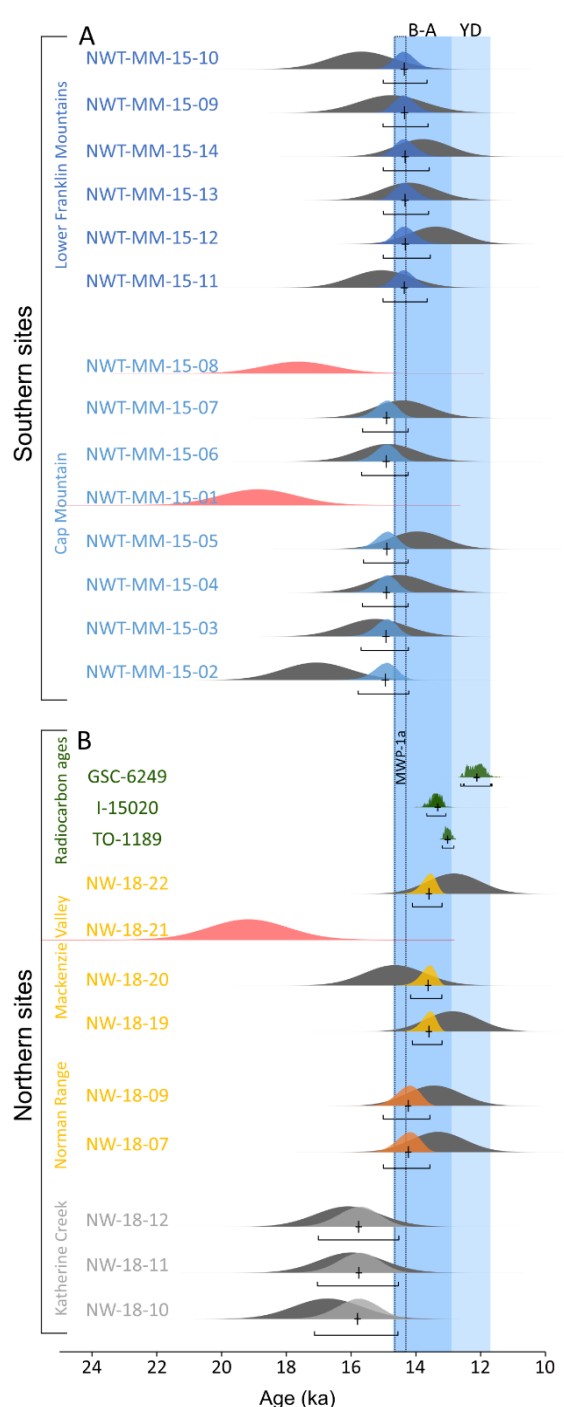

**Figure 6: (A) Phase model from the Bayesian modelling showing the exposure age probability for the southern sites. The dark grey probability distribution plots relate to the age distribution of individual TCN exposure ages. The coloured plots indicate the modelled age probability distribution from our Bayesian modelling. Black crosses indicate the median modelled age. The orange hashed plots indicate samples which were down-weighted within the model. Full information on the Bayesian modelling is provided in figure S3 and S4 and table S4 and S5. (B) Phase model from the Bayesian modelling showing the exposure age probability for the northern sites. The red plots indicate ages which were identified as outliers within the model. The probability distribution of the three oldest radiocarbon ages which constrain our model are displayed in dark green. (C) TCN $^{10}$Be age plotted against boulder elevation. The data points are colour coded by site and relate to panels A and B.**

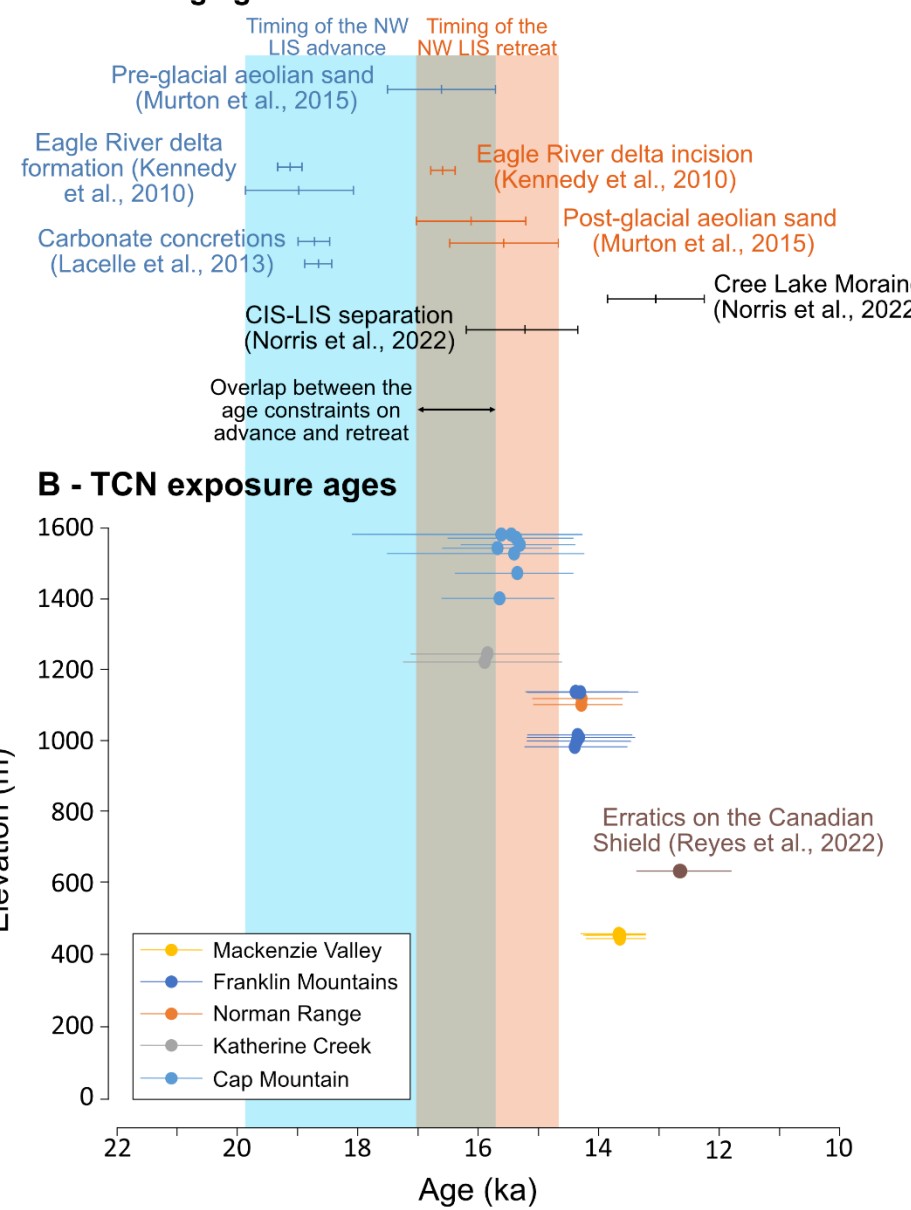

**Figure 7: Comparison of our reconstruction of the timing of deglaciation based on the Bayesian model outputs with existing age constraints. The shaded orange area indicates timing of retreat of the NW LIS including dating method uncertainties and the shaded**

**blue area indicates the timing of advance of the NW LIS including dating method uncertainties based on the age constraints in Panel A. (A) Existing age constraints on the timing of NW LIS advance and retreat to the LGM. The chronological constraints of the advance of the NW LIS to the LGM are shown as blue points and the constraints on the timing of retreat from the LGM are shown in orange. The timing of deglacial events from the multi-chronometer Bayesian model reconstruction of the SW LIS is shown as black points (Norris et al., 2022). Points in this panel are not placed in an elevation sequence. (B) Age-elevation plot using the**

**modelled age distribution output of Bayesian model 2 and 5 (Fig. S3 and S4; Table S3 and S4). The exposure ages were calculated using the 'primary' calibration dataset (Borchers et al., 2015), LSDn scaling, and a correction for glacioisostatic adjustment (as described in Section 2.1.3).**




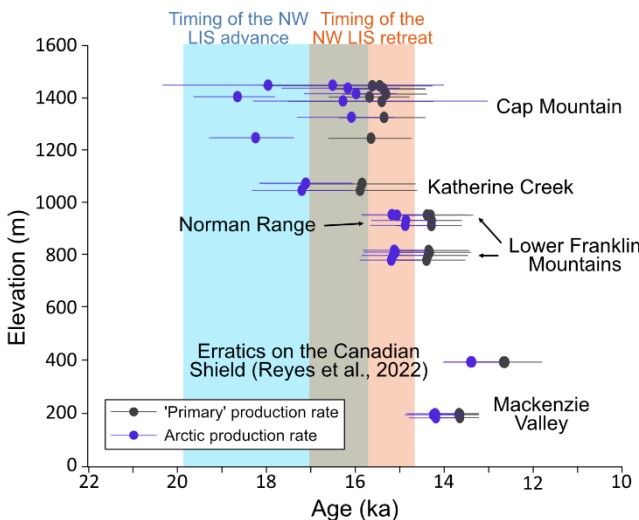

**Figure 8: The influence exposure age calculation approach on the reconstructed age of deglaciation. Shaded orange and blue areas**
**are based on the age constraints in Figure 7. The 'primary' production rate exposure ages were calculated using the 'primary'**
**calibration dataset (Borchers et al., 2015), LSDn scaling, and a correction for glacioisostatic adjustment (GIA) (as described in**
**Section 2.1.3) and plotted using the modelled age distribution output of using the modelled age distribution output of Bayesian model**
**2 and 5 (Fig. S3 and S4; Table S3 and S4). The Arctic production rate exposure ages were calculated using the Arctic production**
**rate (Young et al., 2013), Lal/Stone scaling, and a correction for GIA (as described in Section 2.1.3) and plotted using the modelled**
**age distribution output of Bayesian model 8 and 11 (Fig. S5 and S6; Table S5 and S6). The wide spread of modelled ages for the Cap**
**Mountain site is a result of the smaller uncertainties associated with exposure ages calculated with the Arctic production rate.**

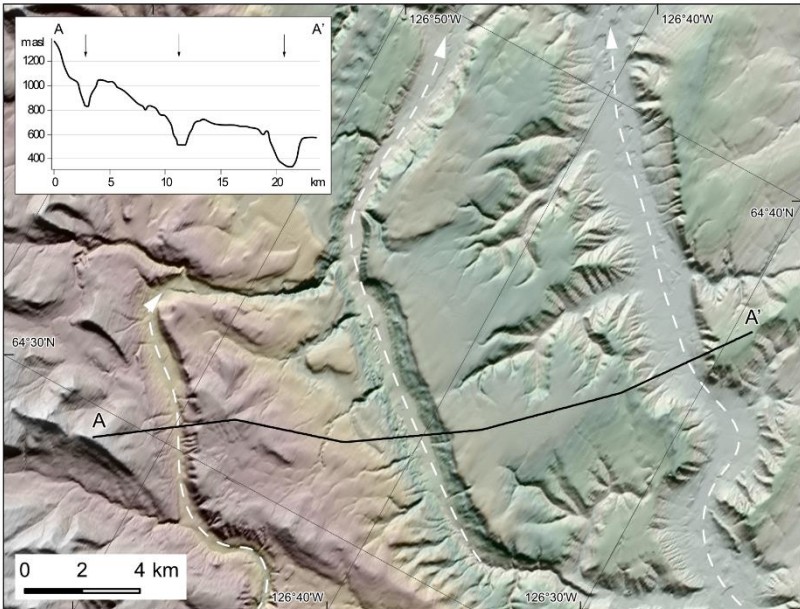

**Figure 9: A series of large marginal spillways along the eastern slopes of the Mackenzie Mountains. White dashed arrows indicate**
**the meltwater flow direction. The location of this figure is indicated on Figure 2.**





**Table 1. Cosmogenic ¹⁰Be sample data and modelled surface exposure ages.**

| Sample code | Latitude / Longitude (°N/°W) | Elevation (m above sea level) | Uplift correction (m) | Sample thickness[a] (cm) | Height of boulder above ground[b] (m) | ¹⁰Be Concentration[c] (x10³ atoms g⁻¹ SiO₂) | Age[c, d] zero erosion (ka) | Age[c, d] GIA corrected (ka) | Age[c, d] Arctic production rate[e] (ka) |
|---|---|---|---|---|---|---|---|---|---|
| NWT-MM-15-01 | 63.39880 / 123.18618 | 1398 | 33.2 | 4 | 0.6 | 301.77 ± 8.92 | 18.6 ± 1.2 | 19.2 ± 1.3 | 20.7 ± 1.0 |
| NWT-MM-15-02 | 63.40141 / 123.18961 | 1441 | 33.2 | 3 | 0.2 | 283.87 ± 7.77 | 16.7 ± 1.1 | 17.3 ± 1.1 | 18.6 ± 0.9 |
| NWT-MM-15-03 | 63.40141 / 123.18961 | 1441 | 33.2 | 2.5 | 0.2 | 252.80 ± 8.16 | 14.8 ± 1.0 | 15.3 ± 1.0 | 16.5 ± 0.8 |
| NWT-MM-15-04 | 63.40084 / 123.18895 | 1430 | 33.2 | 2.5 | 0.35 | 238.02 ± 7.58 | 14.1 ± 0.9 | 14.6 ± 1.0 | 15.7 ± 0.8 |
| NWT-MM-15-05 | 63.39967 / 123.18566 | 1409 | 33.2 | 2.5 | 1 | 223.88 ± 5.94 | 13.5 ± 0.9 | 13.9 ± 0.9 | 15.0 ± 0.7 |
| NWT-MM-15-06 | 63.39743 / 123.18257 | 1381 | 33.2 | 2.5 | 0.6 | 233.90 ± 6.41 | 14.4 ± 0.9 | 14.9 ± 1.0 | 16.0 ± 0.7 |
| NWT-MM-15-07 | 63.39445 / 123.17811 | 1320 | 33.2 | 2 | 1 | 215.48 ± 5.88 | 14.0 ± 0.9 | 14.4 ± 0.9 | 15.5 ± 0.7 |
| NWT-MM-15-08 | 63.38852 / 123.16958 | 1241 | 33.2 | 1.5 | 0.4 | 250.43 ± 6.86 | 17.3 ± 1.1 | 17.9 ± 1.2 | 19.1 ± 0.9 |
| NWT-MM-15-09 | 63.39482 / 123.13284 | 794 | 33.2 | 2.5 | 0.4 | 138.38 ± 6.14 | 14.3 ± 1.1 | 14.9 ± 1.1 | 15.7 ± 0.9 |
| NWT-MM-15-10 | 63.39434 / 123.13240 | 776 | 33.2 | 1.5 | 0.5 | 146.45 ± 4.70 | 15.3 ± 1.0 | 15.8 ± 1.1 | 16.8 ± 0.8 |
| NWT-MM-15-11 | 63.21141 / 123.13831 | 949 | 33.2 | 2.75 | 0.4 | 162.34 ± 4.53 | 14.7 ± 1.0 | 15.2 ± 1.0 | 16.1 ± 0.8 |
| NWT-MM-15-12 | 63.21064 / 123.13758 | 947 | 33.2 | 2 | 0.5 | 143.68 ± 4.05 | 12.9 ± 0.8 | 13.4 ± 0.9 | 14.2 ± 0.7 |
| NWT-MM-15-13 | 63.06287 / 122.99075 | 816 | 33.2 | 3 | 0.7 | 134.68 ± 3.68 | 13.7 ± 0.9 | 14.2 ± 0.9 | 15.1 ± 0.7 |
| NWT-MM-15-14 | 63.06110 / 122.99104 | 809 | 33.2 | 3 | 1.2 | 129.50 ± 3.53 | 13.3 ± 0.9 | 13.8 ± 0.9 | 14.6 ± 0.7 |
| NW-18-07 | 65.24351/ 126.10814 | 943 | 10.8 | 2 | 0.6 | 140.33 ± 2.98 | 13.2 ± 1.1 | 13.3 ± 1.1 | 14.2 ± 0.6 |
| NW-18-09 | 65.24564 / 126.10814 | 924 | 10.8 | 2 | 1 | 139.22 ± 3.02 | 13.3 ± 1.1 | 13.4 ± 1.1 | 14.4 ± 0.6 |
| NW-18-10 | 64.98669 / 127.58993 | 1048 | 1.1 | 1 | 0.4 | 196.29 ± 4.14 | 16.7 ± 1.4 | 16.7 ± 1.4 | 17.9 ± 0.8 |
| NW-18-11 | 64.990309 / 127.58282 | 1075 | 1.1 | 4 | 0.5 | 187.79 ± 4.04 | 16.0 ± 1.4 | 16.0 ± 1.4 | 17.1 ± 0.7 |
| NW-18-12 | 64.990361 / 127.57427 | 1071 | 1.1 | 2 | 0.6 | 191.42 ± 4.09 | 16.1 ± 1.4 | 16.1 ± 1.4 | 17.3 ± 0.7 |

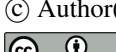



| | | | | | | | | |
|---|---|---|---|---|---|---|---|---|
| NW-18-15 | 63.85554 / 126.87129 | 1394 | NA | 1.5 | 1 | 491.94 ± 10.58 | 31.4 ± 2.7 | NA | NA |
| NW-18-16 | 63.855578 / 126.86933 | 1385 | NA | 1.5 | 0.7 | 1498.12 ± 28.31 | 97.9 ± 8,1 | NA | NA |
| NW-18-17 | 63.856929 / 126.86648 | 1374 | NA | 2.5 | 1.4 | 368.70 ± 7.27 | 24.1 ± 2.0 | NA | NA |
| NW-18-18 | 63.856685 / 126.85992 | 1374 | NA | 1.5 | 1 | 3718.45 ± 51.70 | 254.9 ± 20.3 | NA | NA |
| NW-18-19 | 65.18500 / 126.29600 | 192 | 10.8 | 5 | 0.3 | 65.30 ± 1.60 | 12.7 ± 1.1 | 12.9 ± 1.1 | 13.9 ± 0.6 |
| NW-18-20 | 65.186426 / 126.29890 | 203 | 10.8 | 1.5 | 0.6 | 77.24 ±1.88 | 14.5 ± 1.3 | 14.7 ± 1.3 | 15.8 ± 0.7 |
| NW-18-21 | 65.186608 / 126.29932 | 207 | 10.8 | 2 | 0.4 | 100.95 ± 2.14 | 18.7 ± 1.6 | 19.2 ± 1.6 | 20.7 ± 0.9 |
| NW-18-22 | 65.186495 / 126.29866 | 204 | 10.8 | 1.5 | 0.2 | 67.72 ± 1.65 | 12.7 ± 1.1 | 12.8 ± 1.1 | 13.9 ± 0.6 |
| AVR16-04* | 65.1049 / 115.720 | 470 | 76.6 | 2 | N/A | 83.21 ± 3.03 | 12.0 ± 0.8 | 12.9 ± 0.9 | 13.8 ± 0.7 |
| AVR16-05* | 65.1049 / 115.719 | 470 | 76.6 | 2 | N/A | 80.00 ± 2.94 | 11.5 ± 0.8 | 12.4 ± 0.9 | 13.3 ± 0.7 |
| AVR16-06* | 65.1044 / 115.719 | 470 | 76.6 | 2 | N/A | 78.57 ± 3.63 | 11.3 ± 0.8 | 12.2 ± 0.9 | 13.0 ± 0.8 |
| AVR16-07* | 65.1044 / 115.718 | 470 | 76.6 | 2 | N/A | 81.85 ± 3.66 | 11.8 ± 0.9 | 12.7 ± 0.9 | 13.6 ± 0.8 |
| AVR16-08* | 65.1044 / 115.718 | 470 | 76.6 | 2 | N/A | 78.80 ± 2.52 | 11.3 ± 0.8 | 12.2 ± 0.8 | 13.0 ± 0.6 |
| AVR16-09* | 65.1040 / 115.718 | 470 | 76.6 | 2 | N/A | 80.85 ± 3.73 | 11.6 ± 0.9 | 12.5 ± 0.9 | 13.4 ± 0.8 |
| JR16-224* | 65.2703 / 113.270 | 400 | 62.4 | 2 | N/A | 71.32 ± 2.76 | 11.0 ± 0.8 | 11.7 ± 0.8 | 12.5 ± 0.7 |
| JR16-225* | 65.2703 / 113.270 | 400 | 62.4 | 2 | N/A | 78.88 ± 2.53 | 10.4 ± 0.7 | 11.1 ± 0.8 | 11.9 ± 0.6 |
| JR16-226* | 65.2695 / 113.272 | 400 | 62.4 | 2 | N/A | 42.85 ± 12.91 | 6.6 ± 2.0 | 7.0 ± 2.2 | 7.5 ± 2.3 |
| JR16-227* | 65.2711 / 113.274 | 400 | 62.4 | 2 | N/A | 67.60 ± 3.45 | 10.3 ± 0.8 | 11.0 ± 0.9 | 11.9 ± 0.8 |
| JR16-228* | 65.2719 / 113.273 | 400 | 62.4 | 2 | N/A | 65.47 ± 3.03 | 10.0 ± 0.8 | 10.7 ± 0.8 | 11.5 ± 0.7 |
| JR16-229* | 65.2720 / 113.273 | 400 | 62.4 | 2 | N/A | 67.39 ± 1.60 | 10.3 ± 0.7 | 11.0 ± 0.7 | 11.9 ± 0.5 |

[a] The tops of all samples were exposed at the surface.

[b] Shielding factor is 1 for all samples.

[c] All uncertainties are reported at the 1-sigma level. Blank corrected 10/9 ratios. See the Methods section for details on level of blanks.



[d] Exposure ages were calculated with the online calculator formerly known as CRONUS (Balco et al., 2008); version 3.0; constants 3.0.4
(http://hess.ess.washington.edu/). Full details of the cosmogenic $^{10}$Be analyses and exposure age calculations are provided in the Methods section.

[e] Exposure ages calculated following the approach set out in Berto et al (2022), using the Arctic production rate and the Lal/Stone scaling method. A correction
    for GIA was made following the approach outlined in the Methods section.

*Ages previously published in Reyes et al. (2022).
