# Peer review of "The collapse of the Laurentide-Cordilleran ice saddle and early opening of the Mackenzie Valley, Northwest Territories, Canada, constrained by 10Be exposure dating"

_The Cryosphere, 2022_

## Author Response (AR1)

**Editor comments:**

I would like to thank both reviewers for their thorough and constructive comments on this manuscript and also the authors for posting their response to the reviewers' comments.

Both reviews are positive, but they raise a number of issues that require clarification or revision to the original manuscript. In their response, the authors clearly outline the steps that they propose to take to address all the points raised by the reviewers. I therefore invite them to submit a revised version of their manuscript.

When preparing this revised manuscript, the authors should also reflect on any issues raised in the initial editor's access review. In particular, please check that the choice of model used to apply the glacial isostatic adjustment correction is rigorously justified and the method employed to make this correction is clear. Also, when reviewing the Discussion sections, check that arguments are rigorously justified and concisely articulated.

Thank you for your time in editing our manuscript and the useful comments you have provided. We have pasted the initial editor's access review below and reply to any issues raised in red text.

ORIGINALITY / NOVELTY (1-4): 2

This study combines new terrestrial cosmogenic nuclide (TCN) analysis with existing TCN-, radiocarbon- and OSL-dated samples from the NW sector of the former Laurentide Ice Sheet to reconstruct the deglacial history of this region following the last glacial maximum. Results are compared with ice sheet model output to estimate the sea-level contribution associated with the ice loss. The study fits the remit of the journal in that it provides new constraints on the past evolution of one the main global ice sheets. The results also provide insight into a number of long-standing research questions related to the timing and nature of Laurentide deglaciation. The techniques employed are not new, but the authors provide useful discussion of the impact of choices made when processing/analysing the data, and they draw on comparisons with previous work to identify best practice.

SCIENTIFIC QUALITY / RIGOR (1-4): 2/3

The methods for sample preparation and analysis are clearly documented but some aspects of the exposure age calculation are unclear and there is a lack of detail in the description of the comparison between field evidence and ice sheet model output.

Thank you for the comment. We believe the edits made in relation to the reviewer comments have clarified and improved the exposure age calculation description in the main text and we outline below the method for comparing between field evidence and ice sheet model output.

The choice of model used to apply the glacial isostatic adjustment correction could have been more rigorously justified (the three models considered are very different, fig. S1)

The model choice was predominantly based upon model resolution. As our study area is located at the fringes of the Laurentide Ice Sheet, the higher resolution of the Lambeck et al. (2017) model (0.25° x 0.25° and 0.5 ka timesteps) compared to the models of Gowan et al. (2021) (1° x 1° and 2.5 ka timesteps) and Peltier et al (2015) (1° x 1°), is beneficial. We believe this allows a more accurate representation of the GIA response across our study area. In addition, the region-specific nature of the Lambeck et al. (2017) model means that it is constrained directly by empirical data from our study region (the tilt of shorelines from the former glacial lakes Mackenzie and McConnell), and we

believe this means the GIA response in this model will be most accurate for our study region. We have now added a further detail in the text to better describe this.

and some aspects of the correction method are a little confusing, e.g. why rely on the model of Lambeck et al. (2017) to identify when the sites became ice free (line 153) if you have exposure age data (albeit uncorrected) that provide the same information? This is something that will hopefully become clear through the review process.

To avoid over-estimating the influence of GIA when calculating TCN exposure ages, it is necessary to use the timing of ice-free conditions in the GIA model rather than our new TCN exposure age constraints. When calculating the ΔRSL value to correct the modern elevation, we are interested in the change in elevation since the site became ice-free and exposed to cosmic rays. Using our TCN exposure age to determine the time a site became ice-free in the GIA model and then calculating the ΔRSL would include periods of uplift which occurred at the site while it was still under ice in the GIA model and not exposed to cosmic rays. This results in an over-estimation of the amount of uplift that occurred since the boulder was exposed to cosmic rays.

The quantification of the sea-level contribution does not appear to directly rely on the new TCN data and hence this aspect of the study is somewhat speculative.

The sea-level contribution calculation is based on the simulations presented in Gregoire et al. (2016). In Gregoire et al. (2016), they identify 25 'Not Ruled Out' simulations in the CANO ensemble which they deem as realistic based on the reconstruction of Dyke (2004). Using our new reconstruction, we apply a further constraint on these simulations that the Franklin Mountains should not have deglaciated prior to 16ka (based on our new TCN exposure ages). We use the remaining 'Not Ruled Out' simulations to calculate our 'average' sea level rise contribution that we present in the text. While we do not provide any new model simulations within this study, we believe that the use of our new reconstruction allows a more accurate sea level rise contribution by ruling out the simulations which do not match the new empirical data. We have updated the text to better highlight the contribution we make in this regard.

SIGNIFICANCE / IMPACT (1-4): 2

The study provides new constraints on the timing of ice thinning in the NW sector of the former Laurentide Ice Sheet. The results disagree with some previous work, prompting a useful discussion of the different approaches to exposure age dating. The new findings are used to motivate a revision to the pattern and timing of ice retreat in the region. This revised ice history has implications for understanding migration routes and meltwater flux to the ocean but some of the wording when discussing these issues could be a little more cautious.

PRESENTATION QUALITY (1-4): 2

The manuscript is generally well organised and well written but some sections of the text could be more concise. Figures are informative and of good quality, but a number of features are not labelled or are unclear. For example, in figure 2 it is difficult to identify the various ice sheet margins, particularly in the south-west;

We have made the margin lines more visible.

figures 3 and 9 are missing a colour scale. In revised versions, please ensure all locations mentioned in the text are labelled on a figure, labels would be particularly useful on figure 4. Figure 6C appears to be missing.

These changes have been made in response to reviewer comments.

**Reviewer 1:**

**General comments**

This is a well-crafted paper that presents new insights on the deglacial chronology of a vast area of the NW LIS and is constrained by a significant number of new surface exposure dates. The paper addresses relevant scientific questions within the scope of TC and is of high scientific quality, and the results are well supported using thorough methods and modelling. Substantial conclusions are reached – namely the authors propose meaningful modifications to ice margin retreat positions in an area clearly lacking dated material, and the results have a significant impact on ice retreat dynamics, the timing of ice-saddle collapse, meltwater delivery to the Arctic Ocean and contribution to sea level rise, and on the idea of an early opening of the IFC for peopling of North America.

The authors give proper credit to related work and clearly indicate their own original contribution. Model outputs and inputs, sensitivity analysis results for different GIA models, and description of model calculations provided in the text and Supplementary materials are complete and precise to allow their reproduction by fellow scientists. The overall presentation is well structured and clear; language is fluent and precise. The number and quality of figures is appropriate. Figure 6C is missing however. The references are generally appropriate. Several references cited in the text or in the figures are missing in the list, and some listed references do not appear in the text.

Overall a very good contribution that requires minor revisions outlined in the comments below.  Specifically minor parts of the text and figures should be clarified and/or modified. Technical corrections are required or highly recommended.

Thank you for taking the time to review our manuscript and for the detailed comments. We copy your comments below in black and respond to each of them in red text. When we copy text from the manuscript into the response we 'underline' sections where we have added new text and 'strikethrough' to indicate deleted text.

**Specific comments:**

45: Clarify timing of persistent IFC from the earlier models. During entire last glaciation?

Yes, this is correct. We have now amended this sentence to:

'At the same time, earlier models suggested a lack of coalescence between the LIS and CIS (Johnston, 1933; Antevs, 1935; Mandryk, 1996), and thus a persistent IFC between these ice sheets during the last glaciation.'

73: Boulders suited for TCN analysis should be large in size (>1 m) and stable. Some boulders from this study appear rather small and/or considerably embedded in sediments on Figure S2. Explain reasoning for choosing such small boulders and if exposure ages may have been affected by movement or exhumation – in methods section or caption of Figure S2.

We have now expanded upon the methods section to highlight the potential issues of sampling smaller boulders. It now reads:

'In particular, we preferentially sampled the surface of erratic boulders which: were situated on stable ground away from steep slopes (Heyman et al., 2011), display a rounded shape suggesting a longer transport history by the ice sheet, exhibited limited evidence of surface weathering (Balco, 2011), and were large and well exposed above the ground surface (Heyman et al., 2016). Sampled boulders for TCN exposure dating should be exposed >1m above the ground where possible. Limited

boulder availability meant that some smaller boulders were sampled. These smaller boulders are more likely to be impacted by shielding from snow cover or to have been exhumed following the denudation of surface till cover, resulting in an exposure age which underestimates the true deglaciation age (Heyman et al., 2016). In our sampling area, the annual precipitation rates are very low (average snow depth of less than 30cm; Government of Canada, 2019), meaning that snow cover is unlikely to be an issue. The majority of our samples were taken from high elevation sites with very limited till cover, meaning the sampled boulders are unlikely have been covered in till after deposition.'

Below, we provide two charts showing sample exposure age against boulder height. In these charts, there is no clear relationship between boulder height and exposure age which would suggest that the exposure age was influenced by snow cover or denudation of a surface till layer. We will include these charts in the supplementary material.

[Figure]

[Figure]

134-135: Provide range of % change of the minimal impact of changes in atmospheric mass distribution on the exposure age from cited studies.

In the cited studies, the impact of atmospheric mass distribution changes on the calculated exposure ages is over an order of magnitude less than that of GIA impacts. In Cuzzone et al. (2016), the calculation for atmospheric changes was about 4% of the GIA correction, resulting in ages ~1% younger. In Ullman et al. (2016), the atmospheric correction ranged from 1 – 5 % of the GIA correction for their sample sites. In Dulfer et al. (2021), the atmospheric correction was ~1% of the GIA correction. We now include these values in the main text.

299 and Fig. 8: The two clusters of ages at the Cap Mountain summit site are not obvious on Fig. 8 and S6. They appear more like a spread. Clarify.

Thank you for the comment. You are correct that spread is a more appropriate term to describe the exposure age distribution. We have now changed the description to 'spread' instead of 'cluster' when referring to this site.

308: Is the change in the retreat pattern and style provided by the new TCN ages also supported by ice marginal landforms and/or changes in the glacial erosional record?

There is geomorphological evidence in the record of glacial lineations of the shift from SE-NW ice flow to a more E-W oriented ice flow in the central Mackenzie Valley region around the Norman Range and Franklin Mountains. Currently, we have a Journal of Maps publication in review which documents this geomorphic evidence. Surficial maps from the region also document these changes in ice flow, but the past ice flow dynamics have not yet been properly reconstructed for the NW LIS. Currently, our ice margin reconstruction is drawn based purely on tracing the elevation from our TCN exposure dating sites and drawing glaciologically plausible ice margins. We accept that this leads to a very simplified ice margin reconstruction that may not be completely accurate, but we believe this is appropriate at the broad-scale of our ice margin reconstructions. A higher resolution

reconstruction of the ice margin retreat and past changes in ice flow based on empirical geomorphological data is planned in a future manuscript.

335: Provide reference for given radiocarbon dates (13.4 ± 0.17 cal ka BP).

Done.

447: It is difficult to see the ice lowering of 116-157 m versus 19-47 m on Fig. 5A. Can you point to these drops on the figure?

We have now colour coded the model simulations and indicated the time periods we refer to using a shaded area. We believe it now clearly shows that an early saddle collapse, during MWP-1a, only occurs in the model simulations where surface lowering starts before the Bølling-Allerød.

478-481: Consider adding underlined text for clarification: … is an earlier retreat of the NW LIS by about 1000 years around 63N in the central Mackenzie Mountains…

Done.

Figure 2: Outline of Fig. 7 represents Fig. 9; clarify if ages besides the red outlines represent mean values of various samples; grey lines difficult to differentiate from drainage; also where does the LGM (21.1 cal) limit appear – difference with "local LGM limit at 18.0 cal"?; consider adding "(two sigma range)" after "deglaciation ages".

We have made the suggested changes.

The ages beside the red outlines are the three oldest ages within the cluster of ages. We chose to group the dates within the red outlines due to the high density of ages in some areas, which makes the map very cluttered if we present all the ages. We now clarify this in the figure caption.

The ice margin pattern and chronology in this sector of the ice sheet are not very well constrained in the past reconstructions. The ice margin is in a very similar position at 21.1 cal. ka and at 18.0 cal. ka, but do not align very well. This means that including the 21.1 cal. ka margin to this figure results in overlap with the existing margins and a very confusing, cluttered figure. We chose the margin timesteps presented in Figure 2 as they cover the full time period from the local maximum ice sheet extent to complete deglaciation, but are easily interpreted in a single figure panel. We have provided an example of the figure below including the 21.1 cal. ka margin as a solid black line below.

[Figure]

Figure 3: Consider adding reference for ArcticDEM from Polar Geospatial Center.

Done.

Figure 4: Consider indicating location of Mackenzie Valley or River and locations of 6 studied sites.

Done.

Figure 5: Mention what are the red dots as well.

The red dots indicate our southern sampling site for TCN exposure dating in the Franklin Mountains.

Figure 6: The 10Be versus elevation plot is entirely missing in C; add what are B-A and YD and the two vertical blue dashed lines.

Done.

Figure 7: Replace "points" by "lines" in caption of (A).

Done.

Figure 8: Delete one of the repeated "using the modelled age distribution output of" in caption.

Done.

Supplementary Document #1

- 14.2 cal ka: In fact, the ice margin north of ~65°N does not replicate that of Dalton et al. (2020). Explain the slight changes proposed.

This was a typo. It should read '… north of ~66°N…' We have now changed this. The changes around 65°N are justified by the TCN exposure ages from the Mackenzie Valley and Norman Range, with an ice margin then extrapolated north, based on the local topography.

- 13.5 cal ka: What is the basis for the changes of the ice margin position north of 65°N? Geomorphology, elevation, TCN dates?

We have drawn the ice margin position to the north of 65°N based on our TCN exposure dates and then tracing an ice margin along the local topography. This results in the LIS being constrained to the Great Bear Basin at this time. We have rewritten the text to better describe this.

**Technical corrections:**

46-47: In addition to what? There needs to be a link to previous sentence here.

We have removed these connecting words.

60: Although Consider replacing by "However".

Done.

62: Reference should be Dyke et al 2003 (as in list).

Done.

64: Consider adding "(Fig. 2)" after 65°N.

Done.

137: effect

Done.

142: Missing Jones et al., 2017 in reference list. Unless it should be Jones et al. 2019.

You are correct, it should be Jones et al. 2019. We have now corrected this.

162-163: Missing Small et al., 2020 in reference list.

This should have been Small et al. 2019. We have now corrected this.

184: Consider referring to Table S2 at the end of sentence.

Done.

186: Should be Table S2 and not S3 here.

Done.

190: Consider adding "at the Summer Island site" after located.

Done.

201: Consider adding "we" in …and so we do not refer…

Done.

211: Move bracket before "see" (see Gregoire et al., 2016)

Done.

284: Add "was" after Mackenzie Valley floor.

Done.

305: Add reference to Fig. 2 at end of sentence.

Done.

306: Figure 4 does not appear in order as cited in the text. First time this figure is mentioned.

We have removed the earlier mention of Figure 5 from the 'Methods' section and moved this citation to the 'Results' section as this is more appropriate. All figures now appear in order.

388-89-90: Rephrase – difficult to grasp.

We have rephrased this to:

' Using the 'primary' production rate (Borchers et al., 2016) produces mean site ages which are ~8% younger than mean site ages calculated using the Arctic production rate.  The  deglacial chronology calculated using the 'primary' production rate  is more compatible with the existing geochronological constraints (Fig. 7 and 8).'

393: Consider adding "(Fig. 6)" at the end of the sentence (2016).

Done.

453: Add "in" after deglaciation.

Done.

460: Missing (Menounos et al., 2017; Corbett et al., 2019) references in list.

Added.

463: Missing Ivanovic et al 2017 and 2018 references in list.

Added.

465: in

Deleted.

521-22: Delete repeated reference.

Done.

528-29: Reference not found in text.

Deleted.

562: After Barbett, P.J., consider adding "and others".

Done.

622-23: Reference not found in text.

We have added the reference to the text.

660-663: Reference not found in text.

We have added the reference to the text.

708-709: Delete repeated reference.

Done.

739: Incomplete reference?

This is the suggested citation provided by the Geological Survey of Canada in the original document.

Table 1: Missing Berto et al (2022) reference in list.

Done.

Figure S1: Add references for each named model in caption: Peltier et al., 2015; Lambeck et al., 2017; Gowan et al., 2021.

Done.

Figure S2: Provide size of chisel(s).

Done.

Figure S3: Borchers et al., 2015 is given as 2016 in list of references. Verify.

Borchers et al. 2016 is the correct citation. We have amended this throughout the text.

Table S1: Provide Peltier's reference instead of ICE-6G in the headers (to be consistent with others); add "age" to all headers.

Done.

**Reviewer 2:**

The paper by Stoker et al. addresses the timing of continental glaciation in northwestern Canada during the last deglacial period using surface exposure dating and modeling results.

Overall, I find this paper to be well written and provides a fairly comprehensive assessment of the glacial history, prior data, and prior work on the ice history in this region. The data itself are a nice contribution to the existing data and I think warranted for publication. To me, the authors have done a nice job in both presenting their new work and putting it into the context of existing datasets. Please see my line comments for mostly minor suggestions. Some of the questions I have regarding the Bayesian modeling are a little more major as are the reporting of information regarding the 10Be ages, calculations, and lab procedures.

Thank you for taking the time to review our manuscript and for the detailed comments. We copy your comments below in black and respond to each of them in red text. When we copy text from the manuscript into the response we 'underline' sections where we have added new text and 'strikethrough' to indicate deleted text.

Line Comments:

Line 53: Early 36Cl dates - how many dates were presented in the earlier work? Perhaps say.

Done.

Line 59: "On the all-time" – this is colloquial and hard to understand what is meant. Consider rephrasing.

Done.

Line 65: "Dipstick" – define this for the reader.

We have removed the reference to the dipstick approach in this section and we describe our sampling site selection in the methods section.

Line 79: "True" – what is meant by true? Can you define this better for the reader?

We have removed this phrase to prevent any confusion.

Line 98: HF – include the percentage since presumably not concentrated.

We have rewritten this to:

'… etched in a solution of 2:1 concentrated ACS-grade HF acid to deionized water'.

Line 103: The carrier concentration needs to be included for replication purposes. Very important.

Done. It now reads:

'of Be from $BeCl_2$ carrier "*Be Carrier B31 Sept 28, 2012*" which was produced at CRISDal from phenacite sourced from the Ural Mountains, with an ICP-OES-measured average Be concentration of 282 ± 5.64 µg/ml (replicated by N. Lifton at PRIME Lab with a measurement of 279 µg/ml) and density of 1.013 g/ml. The $^{10}Be/^9Be$ of the carrier was less than 1 x $10^{-16}$.'

Line 112: The blank values need to be stated.  I suggest included them in both the text and supplemental to help the reader understand the precision of the measurements.

Done, this now reads:

'The process blanks had $^{10}$Be/$^{9}$Be ranging from 2.7 to 8.2 x 10$^{-16}$, so for all samples this correction was less than 1% of the adjusted $^{10}$Be values.'

Lines 117-118: Again, what are all these uncertainties?  Please state them (%) to help the reader understand how each is incorporated in the measurements.

Done. This now reads:

'Individual ages are reported to three significant figures with a 1σ external error (Balco et al., 2008) which considers systematic uncertainties in site production rate,  and internal error which includes the following random error sources, added in quadrature: (i) 1σ AMS precision in $^{10}$Be/$^{9}$Be (atoms/atoms), which averaged 2.1% and is the greater of the Poisson distributed statistic for the total number of counts on a target or the coefficient of variation about the mean $^{10}$Be/$^{9}$Be after three, four, or five passes on each target; (ii) 1σ uncertainty in carrier concentration (µg/ml) which includes uncertainty in density and is based on the greater of the 1-standard deviation of three measurements for a given wavelength or the standard deviation of the two wavelengths (313.042 nm and 234.861 nm). Carrier concentration uncertainty for *Be Carrier B31 Sept 28, 2012* was less than 2% over 9 measurements, but is rounded to 2%; and (iii) 1σ error in sample Be concentration (µg/ml) as measured by ICP-OES, and error contributed by uncertainty in the process blank, which is calculated in the manner as (ii).'

Line 134: What did Cuzzone et al. find?  State what is meant by "minimal".

Reviewer 1 had a similar comment, I am copying the reply here:

'In the cited studies the impact of atmospheric mass distribution changes on the calculated exposure ages is over an order of magnitude less than that of GIA impacts. In Cuzzone et al. (2016), the calculation for atmospheric changes was about 4% of the GIA correction, resulting in ages ~1% younger. In Ullman et al. (2016), the atmospheric correction ranged from 1 – 5 % of the GIA correction for their sample sites. In Dulfer et al. (2021), the atmospheric correction was <1% of the GIA correction. We now include these values in the main text.'

Line 135: Model resolution – what is meant by suitable resolution?  The models that were used in Cuzzone et al. will have the same resolution for this location.  I don't think resolution should factor into any of this or be a justifiable reason to not do it.  There is also statistically downscaled data for North America that could be used, so it might in fact be better than the model used in Cuzzone et al.

The Alder and Hostetler (2015) model used by Cuzzone et al. (2016), Ullman et al. (2016), and Dulfer et al. (2021) has a spatial resolution of 3.75° x 3.68° and a temporal resolution of 3ka. The spatial resolution means that a single pixel will cover all of our six TCN exposure dating sample sites. While the rapid retreat over our study area will have been associated with rapid changes in the atmosphere which will not be represented in the 3 ka timesteps of the model.

The resolution is not the main reason that we do not undertake this step. The previous studies (listed in the response to the comment above) all found that a correction for changes in the atmospheric composition was at least an order of magnitude smaller than a correction for GIA-related changes in elevation. This tends to result in exposure ages that are ~1% older. Therefore, we believe it is not necessary to apply this correction as the method and models have previously been

found not to substantially affect the calculated exposure age. This does not rule out the fact that improved models of atmospheric composition changes and an improved understanding of the processes operating following deglaciation may require that our ages undergo a correction for atmospheric composition changes in future.

Finally, we believe that the impacts of changes in the atmosphere during deglaciation are likely to have been minimal at our study sites. Staiger et al. (2007) identified that the production rate at an ice marginal site could be up 10% different to the modern day production rate. The strength of this influence is dependent on how long a site is exposed to these changed conditions. Staiger et al. (2007) suggested that sites which were adjacent to an ice margin for prolonged periods (e.g. nunataks or sites at the LGM margin will be most strongly influenced by these changed atmospheric conditions. Our study area experienced rapid deglaciation, and none of our sites fit the characteristics outlined above. This rapid retreat meant that our sampling sites were in an ice margin position for a very limited time and were therefore not exposed the changing atmosphere or katabatic winds for long enough to substantially affect the exposure age.

Lines 164-165: How does the spatial scale factor into the Bayesian model and the priors?  It is clear when doing an elevational transect with the 10Be ages but how do the authors account for the 14C ages coming from different locations.  Elevation and simple horizontal retreat of the margin will control the timing and it isn't clear to me how it all fits together.  I realize this is challenging to explain but I do think it worth trying to explain this to the reader.  The supplemental Figures 7 and 8 are just the line code from Oxcal which really isn't that helpful for parsing this all out.  At the moment, I don't think a reader could reconstruct what the authors have done and/or the assumptions being made.

Spatial scale is not a factor in our Bayesian model and the priors. The $^{14}$C ages we include in our Bayesian modelling setup are located in close proximity to our exposure age dating sites and from a stratigraphic setting which means these sites must postdate deglaciation (e.g. postglacial lake sediments). Therefore, these dates are included using the *Before* function as the stratigraphic setting they are located in means that they provide a strong minimum age on deglaciation which our exposure ages must be compatible with. This was not properly described in the text so we have now included further sentences to describe the model set up:

'Our Bayesian model for the northern sites also included radiocarbon dates. These radiocarbon dates are from post-glacial delta and lake sediments which must postdate our exposure age sampling sites and therefore provide a minimum timing of deglaciation. These radiocarbon ages do not follow our age-elevation prior model, but can be placed in order as they are from sites which must have deglaciated after our northern sites.'

Line 158: General comment on the Bayesian modeling – what are the uncertainties being used in the model for both the 10Be and 14C ages?  Presumably, the authors are using the external uncertainties for the 10Be but this isn't made clear in this section. More information would be useful.

This is a good spot. We have now added a sentence to describe the uncertainties, it reads:

'TCN exposure ages were input to the model using the external error while radiocarbon dates were input using the analytical uncertainty reported by the original authors.'

Line 299 and Figure S2: Many of the boulders selected for this study are fairly small and several look like they are being exhumed.  The authors should discuss this either in the methods or results about

the potential implications. Or, alternatively, give the reader some sense if this is problematic or not (e.g. boulder height plotted versus age by location or in general).

Reviewer 1 had a similar comment on boulder height. We have copied across the comment here:

'We have now expanded upon the methods section to highlight the potential issues of sampling smaller boulders. It now reads:

'In particular, we preferentially sampled the surface of erratic boulders which were: situated on stable ground away from steep slopes (Heyman et al., 2011), display a rounded shape which suggests a longer transport history by the ice sheet, exhibited limited evidence of surface weathering (Balco, 2011), large and well exposed above the ground surface (Heyman et al., 2016). Sampled boulders for TCN exposure dating should be exposed >1m above the ground where possible. Limited boulder availability meant that some smaller boulders were sampled. These smaller boulders are more likely to be impacted by shielding from snow cover or to have been exhumed following the denudation of surface till cover, resulting in an exposure age which underestimates the true deglaciation age (Heyman et al., 2016). In our sampling area, the annual precipitation rates are very low (average snow depth of less than 30cm; Government of Canada, 2019), meaning that snow cover is unlikely to be an issue. The majority of our samples were taken from high elevation sites with very limited till cover, meaning these samples are unlikely have been affected by a thick surface till cover.'

Below, we provide two charts showing sample exposure age against boulder height. In these charts, there is not a strong relationship of taller boulders having older exposure ages. We will include these charts in the supplementary material.'

[Figure]

[Figure]

Figure S2: It might be better to break these photos into locations either as new figures or into rows.

The photos are organised in sequence so that each site is together. We have now added labels to each photo to highlight which site each sample relates to.

Line 275: I find this statement a little circular. It assumes the regional deglaciation a priori and then uses it to argue for relatively minor affects. 40km is quite large when considering it in the context of the elevational sampling being done – one may even argue that the elevation changes are quite small themselves, thus top-down elevation dates could in fact be synchronous or even out of order within the uncertainties. I think the authors need to better justify why this is in fact a minor issue – the surface slope of the ice may be a good place to argue and/or the modeling.

Currently, there are no strict best practice guidelines for the grouping of TCN exposure dating samples, which means the procedure can often be subjective with different approaches having positives and negatives. We believe that we have chosen the most appropriate approach considering the basic ice sheet mechanics, past ice sheet reconstructions, and our TCN exposure dates.

The style of deglaciation was dominated by ice sheet thinning in the region. This is demonstrated by our TCN exposure ages which indicate that when the Franklin Mountains and Norman Range at ~800m became ice-free there was still an ice lobe present in the Mackenzie Valley. Based on these observations, we believe the assumption that the higher elevation Cap Mountain site deglaciated before the lower elevation Franklin Mountains sites is reasonable.

The similar position of the sampling sites means that local topographic effects are unlikely to have influenced the pattern of deglaciation between the sampling sites. These sampling sites are all located on the eastern side of the Franklin Mountains, in the direction of the ice retreat compared to the Cap Mountain site. There are no topographic obstacles up-ice that would have impacted the retreat pattern, so the samples were likely exposed at the same time (Mas e Braga, 2021). So in this

situation, we believe that ~600m of elevation difference is more important than the 20 km between each locality sampled (40km between the furthest two samples).

Our Bayesian modelling set up does not substantially skew the mean site deglaciation ages. A simple mean average of the raw TCN exposure ages at this southern site would suggest a site deglaciation age of 15.0 ka for the Cap Mountain site and 14.5 ka for the lower Franklin Mountains site. Whereas mean site deglaciation ages calculated from the Bayesian modelling suggest Cap Mountain deglaciation at 14.9 ka and the Lower Franklin Mountains at 14.3 ka. Within the uncertainties of TCN exposure dating, these differences are negligible. However, the Bayesian modelling setup allows us to better constrain the uncertainties of our TCN samples and to identify outliers in a clear and reproducible way.

For the reasons outlined above, we believe that the Bayesian modelling setup we used is appropriate and based on reasonable assumptions. Throughout the text we have included further details to justify our approach. We appreciate that other users may believe a different approach is more appropriate, so we have provided all the raw data necessary for others to recalculate and use our data as they see fit.

Reference: Mas e Braga, M., Selwyn Jones, R., Newall, J.C., Rogozhina, I., Andersen, J.L., Lifton, N.A. and Stroeven, A.P., 2021. Nunataks as barriers to ice flow: implications for palaeo ice sheet reconstructions. *The Cryosphere*, *15*(10), pp.4929-4947

Line 282: What does vetted mean? Try to be more exact and descriptive for your reader.

In this sentence, 'vetted model' refers to the Bayesian model where we have excluded the outlier TCN exposure date (NW-18-21). We have now cited the associated main text figure and supplementary figure to clarify this.

Lines 358-359: I don't recommend phrasing things this way unless the authors have made some considerable effort reaching out to L. Tarasov or the authors of the Reyes et al. paper. Did the authors in fact contact the authors? Otherwise, it is just calling them in published form. Please reconsider.

Thank you for highlighting this. This was not our intention in this sentence. We have now rewritten this sentence to:

'The GIA correction of Reyes et al. (2022) is based on simulations from Tarasov et al. (2012) . Instead, we apply the GIA correction described in our methods section and following the method of Norris et al. (2022).'

Lines 370-372: While I don't disagree with the authors argument about the Arctic data and it being appropriate (or not). I do think they need to justify things slightly better – the 47 samples may fit the elevation and ages ranges but the spatial changes in the production rate can be quite high because for a variety of reason (e.g. geomagnetic, snow, atmospheric, etc. affects). It would be useful if the authors provided some sense of where these sites are located relative to their own site. Clearly the Reyes et al. paper used the Arctic sites for this very reason so it would help the reader to understand the why the choice for this paper was made as clear as possible.

There is no production rate which perfectly represents the characteristics of our sample sites, but we believe the 'primary' global production rate is the most appropriate (Figure 8). In the global production rate dataset, the highest latitude sample sites are located at ~57°N. This places our sites approximately equal latitudinal distance between the sites of the global production rate and the

Arctic production rate. While our sampling sites may be more similar to the sites of the Arctic production rate in some characteristics compared to the global production rate, we believe that the elevation and age ranges of the Arctic production rate mean it is not appropriate for our study. Therefore, we opt to follow the established practice of using the global production rate where there is no specific local production rate which is appropriate. We accept that there will be some limitations with this choice. In the text, we now provide details on the latitude, elevation, and age of the sites in the Global production rate to allow readers to easily compare it with the Arctic production rate sites.

Regarding snow cover and atmospheric conditions. The sites of the Arctic production rate are located in the coastal areas surrounding Baffin Bay near to contemporary ice masses. Therefore, the atmospheric conditions at the Arctic production rate sites were influenced by the nearby ice sheet for an extended period of time and are likely still influenced by the nearby ice masses. This situation is not representative of our sample sites which experienced rapid deglaciation. In contrast, the global production rate dataset includes some sites from formerly glaciated locations in Scotland. These sites were not situated near an ice sheet margin for an extended and are perhaps more representative of the atmospheric conditions at our sampling sites. The low precipitation and snow depth (< 30 cm average snow depth) observed at our sampling sites are also similar to the global production rate sites which have low snow cover conditions.

Line 460: Corbett et al. 2019 reference is missing. Also, there are several other papers that could be cited here including Koester et al. and Barth et al. from the New England areas where dipsticks have been used to determine the timing of glacial retreat and presumably sea level contributions.

Thank you for the suggestion. We agree this was an oversight and have now included references to both Barth et al (2019) and Koester et al (2021) which refer to rapid ice sheet thinning associated with the Bølling period.

References in general: many references are missing. Please do a thorough review to make sure you have all of them and all extra ones are removed.

We have reviewed the reference list and in-text citations to remove any errors.

Line 488: Data should never be made available on request. Please provide all data in free, online repository following FAIR and journal data standards.

We have made every effort to make all materials and data used in this study freely available in both the main text and associated supplementary materials on Figshare (https://doi.org/10.6084/m9.figshare.20069222.v2). This sentence was intended to cover any oversights or data we may have missed. We agree that this appears misleading though and have now removed this sentence.

Figure 2: Make the text bigger in the figure – hard to read for us oldies. Try to make clear what is new data from this study and what is existing data from others. Some of the radiocarbon ages say "0.0 and null" for the uncertainty. Is this correct and if it is 0.0 make it another decimal and if null, you need to explain what this means.

We have now increased the font size for the radiocarbon ages to increase clarity.

The TCN exposure ages (yellow triangles) are the new data presented by this study.

Thank you for the comment. This was an error which we have now corrected in the revised manuscript.

Figure 3: Are these 14C or 10Be ages? If the latter, which I know they are, make sure to keep the symbols the same as you have in Figure 1. Text is hard to read – consider making larger. It's also hard to know what is transect or not – can you make an inset for the elevational transects that demonstrates the ages with elevation? This would help the reader understand the spatial and elevation data.

We have changed the symbols to match the TCN exposure ages from Figure 1. We have enlarged the text. We have now added elevation transects on panel A and D.

Figure 4: Need some reference to where the data were collected. Maybe plot the points where the ages are from on the map.

 Done.

Figure 5: Label the ages on the surface elevation profiles for the ice sheet. This will help the reader out a bit.

 Done.

Figure 6: Part C is missing from the figure. I am somewhat confused on final results of the Bayesian modeling. It seems to have picked the most improbable sections of the ages at each of the sites. I think this is largely related to the primary assumption that the dates need to be younger with lower elevation. Or, alternatively, the ages are too uncertain or there are inheritance/exhumation issues with the data to correctly apply this assumption for the Bayesian modeling effort. As reported in this figure (e.g. part A), it would mean that the age tails are the most probable age for four of the PDFs which seems very unlikely to me. This should be addressed. Perhaps this gets clarified once we can see figure C.

In previous versions of the manuscript we included Figure 7B as a single panel within Figure 6 (Figure 6C). In the final manuscript, we developed Figure 6C into a full separate figure (Figure 7). We left the caption for Figure 6C in the submission manuscript in error. We have now removed this portion of the text. Apologies for any confusion this may have caused.

At the northern sites, nearby radiocarbon ages are integrated into the Bayesian model as 'before' ages. The results of including these radiocarbon ages means that the modelled site ages are skewed older, towards the age tail of the possible exposure ages. The modelled site deglaciation ages stay within the uncertainty of the TCN exposure dating method. These radiocarbon dates are high quality and have been replicated, therefore, we believe it is important to include them in the Bayesian model. The Bayesian model set up we have used is the only way to quantitatively integrate all the available chronological constraints in this area and we believe is the most appropriate method.

At the southern sites, the lower Franklin Mountains modelled site ages are skewed slightly younger due to the assumption the lower Franklin Mountains site deglaciated after the Cap Mountain site. We believe this is a fair assumption as the lower Franklins Mountains site is located both (1) at a lower elevation and (2) on the east side of the Franklin Mountains ridgelines, which is in the direction of ice retreat. Both of these factors support the fact the lower Franklin Mountains sites should have deglaciated after the Cap Mountain site. In addition to this, the Bayesian model does not substantially alter the ages at this site (as outlined in a previous comment) and the modelled ages remain well within the uncertainty of the exposure ages.

We understand that there is a lot of debate about the most appropriate way to use TCN exposure ages for reconstructing the retreat of former ice masses. We have made every effort to be as

transparent and open in our approach and we provide all of our raw TCN exposure age data so that it can be worked with by other studies and recalculated as others see appropriate.

Figure 9: I'm not sure if this figure is relevant to the main paper.

We believe that this figure provides some important empirical evidence to support the theory of ice sheet saddle collapse in the region and makes the modelling section more convincing.

Table: the authors should provide a data table for input into Cronus (Balco et al. 2008). This will allow the reader to easily replicate their work and if production rates or scaling change, easily adjust those data into the future. This should be standard policy for all work using this or any calculator. The exact input files should be provided. Apologies if I overlooked it.

This is a good point. We now provide the input table for CRONUS in the supplementary material.

---

## Editor Decision (ED1)

**'The collapse of the Laurentide-Cordilleran ice saddle and early opening of the Mackenzie Valley, Northwest Territories, Canada, constrained by [10]Be exposure dating' by Stoker et al.**

The authors have provided a robust response to the issues raised by the reviewers, and they have implemented relevant edits in the revised version of the manuscript. This is a rigorous and well-written piece of work, and I am delighted to confirm that my decision is: 'publish subject to minor revisions (review by the editor)'. A few points require clarification, and these are detailed below (line numbers relate to the track-change version of the article).

Pippa Whitehouse (Editor)
* * *
**Main comment:** the method used to calculate the GIA correction requires clearer justification. Your argument that using the new TCN ages would over-estimate the influence of GIA (mentioned in the author response document) is robust, but it is not clear to me that identifying "when a site became ice-free according to the model of Lambeck et al. (2017)" (line 185, revised manuscript) is a more accurate approach, given the widely differing rebound curves predicted by the three GIA models you consider (Figure S1). Differences between site-specific GIA model predictions of elevation change since deglaciation are typically >100m, translating into GIA corrections that can amount to several kyrs (Table S1). Your arguments for adopting the Lambeck et al. (2017) model are robust, and I am not requesting that you alter the approach you have used to calculate the GIA correction, but given the lack of independent estimates on postglacial rebound in the region, the statement that the effect of GIA is "reasonably well constrained" (line 166, revised manuscript) is not really justified and I recommend considering the following points as you carry out final revisions to the manuscript:

- both methods of determining the GIA correction (use of GIA model output/new TCN ages) contain errors; consider quantifying this or, at least, comment on how well the assumptions in the Lambeck et al. (2017) model agree with the new chronology presented here
- lines 178-180: briefly quantify the differences described here
- lines 186-187: references to 'sea level data' and 'average ΔRSL' are confusing; review the description of the methods used to calculate the GIA correction
- table S1: what does the column labelled 'standard' represent (include units)?

**Minor comments**

lines 44-46: the logic here is awkward, be more explicit that it is no longer assumed that an ice free corridor persisted between the CIS and LIS throughout the last glaciation

line 118/119 and 437/438: text is repeated [only an issue in the track change version]

line 143-144: mention that the impacts of different methodological choices are quantified in the results section (i.e. not just in the supplementary material) and reasons for preferring not to use the Arctic production rate are discussed in section 4.1.2

lines 213-215: text repeats that of lines 197-199

line 338: "The alternate..." – does this refer to calculations using the Arctic production rates?

line 476: Makenzie -> Mackenzie

line 499: insulation -> insolation

line 502: the wording is a little strong and I suggest editing "...mean that we can quantify..." to something like "...allows us to estimate...". Also, review use of the term 'observed' on line 506

line 264/505/fig. 5 caption: do you use 14 or 15 simulations?

lines 519-520: do these ages relate to the time at which the meltwater channels were originally incised, or the period when they contained meltwater?

---

## Author Response (AR2)

**'The collapse of the Laurentide-Cordilleran ice saddle and early opening of the Mackenzie Valley, Northwest Territories, Canada, constrained by 10Be exposure dating' by Stoker et al.**

The authors have provided a robust response to the issues raised by the reviewers, and they have implemented relevant edits in the revised version of the manuscript. This is a rigorous and well written piece of work, and I am delighted to confirm that my decision is: 'publish subject to minor revisions (review by the editor)'. A few points require clarification, and these are detailed below (line numbers relate to the track-change version of the article).

Pippa Whitehouse (Editor)
* * *
**Main comment**: the method used to calculate the GIA correction requires clearer justification. Your argument that using the new TCN ages would over-estimate the influence of GIA (mentioned in the author response document) is robust, but it is not clear to me that identifying "when a site became ice-free according to the model of Lambeck et al. (2017)" (line 185, revised manuscript) is a more accurate approach, given the widely differing rebound curves predicted by the three GIA models you consider (Figure S1). Differences between site-specific GIA model predictions of elevation change since deglaciation are typically >100m, translating into GIA corrections that can amount to several kyrs (Table S1).

Thank you for this comment, it highlights a lack of clarity in our description of the method used for the sensitivity analysis. When calculating the exposure ages for the sensitivity analysis we use the ExPage calculator. This provides a simple tool to quickly compare the GIA rebound curves and the influence of GIA model choice on exposure age calculation. This calculator uses the timing of ice-free conditions for each model when calculating the GIA corrections (i.e. when applying a GIA correction using the Lambeck et al. 2017 model it takes the timing of ice-free conditions from this model and calculates the influence of the rebound following the beginning of ice-free conditions, and when using the GIA rebound of the Gowan et al (2020) model it takes the timing of ice-free conditions from the Gowan reconstruction). The elevation is then calculated for 500 year timesteps (interpolated in the two models with longer timesteps) and calculates the influence of the GIA rebound for each timestep. This allowed us to identify the Lambeck et al. (2017) model as the most appropriate for our use and to compare how different the influence is from different models. Arguably, this is a more sophisticated and 'accurate' approach than our method which simply takes an averaged change in elevation. However, the exposure age calculations of the ExPage calculator uses an older (slightly outdated) value of the geomagnetic field compared to the CRONUS calculator. The use of this 'old' geomagnetic value leads to the systematic overestimation of exposure ages by about 500 years. Therefore, we opt to use the CRONUS calculator for our exposure age calculation and to apply our simplified method of GIA correction. We believe this is valid as our GIA correction does not differ significantly from that of the ExPage calculator (our GIA correction [Table 1] is typically within 100 years of the GIA correction within the ExPage calculator [Table S1], which is a smaller difference than the ~500 year age overestimation). The main difference in our calculated ages vs the calculated ages of the ExPage calculator is a result of this different value for the geomagnetic field. A further sentence has been added in to the text to better describe the methodology of the sensitivity analysis.

Your arguments for adopting the Lambeck et al. (2017) model are robust, and I am not requesting that you alter the approach you have used to calculate the GIA correction, but given the lack of

independent estimates on postglacial rebound in the region, the statement that the effect of GIA is "reasonably well constrained" (line 166, revised manuscript) is not really justified…

This is a good point. We now rephrase this to 'relatively well constrained', in comparison to models of the changes in atmospheric composition following deglaciation. This sentence is intended to transition from our discussion of atmospheric circulation/composition changes to our discussion of GIA related changes, which are comparatively better understood.

…and I recommend considering the following points as you carry out final revisions to the manuscript:

- both methods of determining the GIA correction (use of GIA model output/new TCN ages) contain errors; consider quantifying this or, at least, comment on how well the assumptions in the Lambeck et al. (2017) model agree with the new chronology presented here

The Lambeck et al. (2017) ice sheet reconstruction is based on the assumptions that there was (1) rapid ice retreat during the Bølling–Allerød period and (2) that the Ice-Free Corridor did not open before 13 ka. Both of these constraints are in line with our new chronology. We have now included a sentence to highlight this point, which hopefully helps to demonstrate that the model of Lambeck et al. (2017) is appropriate for what we use it for. We believe that we now are as transparent as possible in our methodology. We have taken steps to provide detailed discussion and justification for all our calculation and correction procedures for our exposure ages. Unfortunately, we are unable to quantify the uncertainties in the corrections we apply beyond what we already do. At the same time, we are not aware of any other study using cosmogenic exposure dating doing that.

- lines 178-180: briefly quantify the differences described here

Done.

- lines 186-187: references to 'sea level data' and 'average ΔRSL' are confusing; review the description of the methods used to calculate the GIA correction

We have amended this text to increase the clarity of the method.

- table S1: what does the column labelled 'standard' represent (include units)?

In Table S1, the standard column refers to exposure ages calculated using the 'standard' calculation approach of the ExPage calculator which applies no corrections (snow cover, GIA, etc) to the exposure ages. We have now amended this to describe what 'standard' involves.

**Minor comments**

lines 44-46: the logic here is awkward, be more explicit that it is no longer assumed that an ice free corridor persisted between the CIS and LIS throughout the last glaciation

Done.

line 118/119 and 437/438: text is repeated [only an issue in the track change version]

This should now be fixed.

line 143-144: mention that the impacts of different methodological choices are quantified in the results section (i.e. not just in the supplementary material) and reasons for preferring not to use the Arctic production rate are discussed in section 4.1.2

Done.

lines 213-215: text repeats that of lines 197-199

We have deleted the second instance of this sentence.

line 338: "The alternate…" – does this refer to calculations using the Arctic production rates?

This is correct, we have now included this in the text.

line 476: Makenzie -> Mackenzie

Done.

line 499: insulation -> insolation

Done.

line 502: the wording is a little strong and I suggest editing "…mean that we can quantify…" to something like "…allows us to estimate…". Also, review use of the term 'observed' on line 506

Done.

line 264/505/fig. 5 caption: do you use 14 or 15 simulations?

Good spot. We have double-checked the simulations in Fig. 5 and in the original data table used to calculate the sea level rise contribution and we use 15 simulations. We have double-checked this is consistent throughout the manuscript.

lines 519-520: do these ages relate to the time at which the meltwater channels were originally incised, or the period when they contained meltwater?

These ages relate to the two ice margins which bracket the location of the meltwater channels. These channels were likely incised over a period of 10s to first 100s of years and since then have not carried any meltwater, so the time of incision and period they contained meltwater is the same event.